# THE GENERATIVE AI PARADOX:
## *"What It Can Create, It May Not Understand"*

**Peter West**[1*] **Ximing Lu**[1,2*] **Nouha Dziri**[2*] **Faeze Brahman**[1,2*] **Linjie Li**[1*]
**Jena D. Hwang**[2] **Liwei Jiang**[1,2] **Jillian Fisher**[1] **Abhilasha Ravichander**[2]
**Khyathi Raghavi Chandu**[2] **Benjamin Newman**[1]
**Pang Wei Koh**[1] **Allyson Ettinger**[2] **Yejin Choi**[1,2]

[1]University of Washington  [2]Allen Institute for Artificial Intelligence
{pawest,linjli}cs.washington.edu
{ximinglu,nouhad,faezeb}allenai.org

## ABSTRACT

The recent wave of generative AI has sparked unprecedented global attention, with both excitement and concern over potentially superhuman levels of artificial intelligence: models now take only seconds to produce outputs that would challenge or exceed the capabilities even of expert humans. At the same time, models still show basic errors in understanding that would not be expected even in non-expert humans. This presents us with an apparent paradox: how do we reconcile seemingly superhuman capabilities with the persistence of errors that few humans would make? In this work, we posit that this tension reflects a divergence in the configuration of intelligence in today's generative models relative to intelligence in humans. Specifically, we propose and test the **Generative AI Paradox** hypothesis: generative models, having been trained directly to reproduce expert-like outputs, acquire generative capabilities that are not contingent upon—and can therefore exceed—their ability to understand those same types of outputs. This contrasts with humans, for whom basic understanding almost always precedes the ability to generate expert-level outputs. We test this hypothesis through controlled experiments analyzing generation vs. understanding in generative models, across both language and image modalities. Our results show that although models can outperform humans in generation, they consistently fall short of human capabilities in measures of understanding, showing weaker correlation between generation and understanding performance, and more brittleness to adversarial inputs. Our findings support the hypothesis that models' generative capability may not be contingent upon understanding capability, and call for caution in interpreting artificial intelligence by analogy to human intelligence.

## 1 INTRODUCTION

*"What I cannot create, I do not understand." – Richard Feynman*

The recent wave of generative AI, from ChatGPT to GPT4 to DALL-E 2/3 to Midjourney, has sparked unprecedented global attention—with equal parts excitement about the expansive potential applications, and deep concern about the dangers of "intelligence[1]" that seems even to exceed that of humans. Indeed, in both language and visual domains, current generative models take only seconds to produce outputs that could challenge experts with years of skill and knowledge, providing compelling motivation for claims that models have surpassed human intelligence (Bubeck et al., 2023; Surameery & Shakor, 2023). At the same time, probing of models' outputs continues to uncover basic errors in understanding that would be unexpected even for non-expert humans (Dziri et al., 2023; Arkoudas, 2023; Qin et al., 2023). This presents us with an apparent paradox: how do we reconcile

---

*First co-authors.

[1]"Intelligence" and "understanding" here refer particularly to demonstrable aspects of models and technology (as in "Artificial Intelligence" or "Natural Language Understanding").

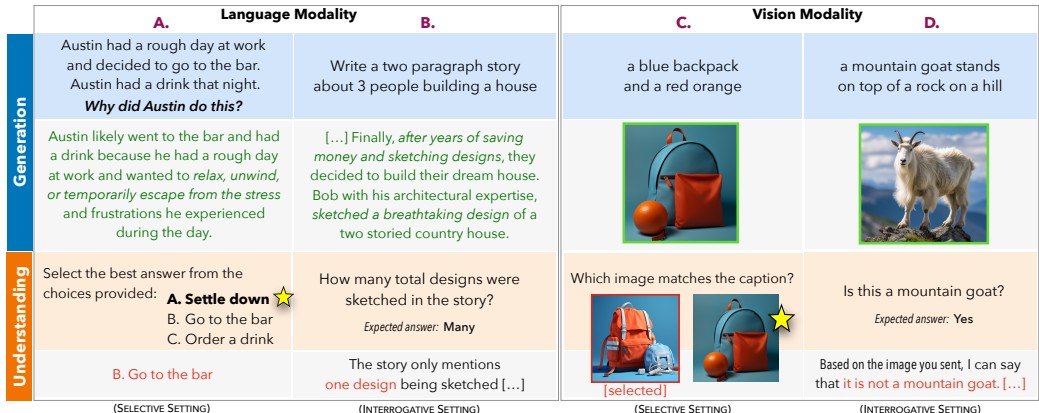

Figure 1: Generative AI in language and vision can produce high-quality generations. Paradoxically, however, models have trouble demonstrating selective (A,C) or interrogative (B,D) understanding of these modalities.

the seemingly superhuman capabilities of these models with the persistent presence of fundamental errors that most humans could correct?

We posit that this tension arises because the configuration of capabilities in today's generative models diverges from the configuration of intelligence in humans. Specifically, in this work we propose and test the **Generative AI Paradox** hypothesis: generative models, having been trained directly to reproduce expert-like outputs, acquire generative capabilities that are not contingent upon—and can therefore exceed—their ability to understand those same types of outputs. This contrasts with humans, for whom basic understanding nearly always serves as a prerequisite to the ability to generate expert-level outputs (Gobet, 2017; Alexander, 2003; Berliner, 1994).

We test this hypothesis through controlled experiments analyzing generation and understanding capabilities in generative models, across language and visual modalities. We conceptualize "understanding" relative to generation via two angles: 1) given a generative task, to what extent can models select correct responses in a discriminative version of that same task? and 2) given a correct generated response, to what extent can models answer questions about the content and appropriateness of that response? This results in two experimental settings, *selective* and *interrogative*, respectively.

Though our results show variation across tasks and modalities, a number of clear trends emerge. In selective evaluation, models often match or even outperform humans on generative task settings, but they fall short of human performance in discriminative (understanding) settings. Further analysis shows that discrimination performance is more tightly linked to generation performance in humans than in GPT4, and human discrimination performance is also more robust to adversarial inputs, with the model-human discrimination gap increasing with task difficulty. Similarly, in interrogative evaluation, though models can generate high-quality outputs across tasks, we observe frequent errors in models' ability to answer questions about those same generations, with model understanding performance again underperforming human understanding. We discuss a number of potential reasons for this divergence in capability configurations for generative models versus humans, including model training objectives, and size and nature of input.

Our findings have a number of broader implications. First, the implication that existing conceptualizations of intelligence, as derived from experience with humans, may not be able to be extrapolated to artificial intelligence—although AI capabilities in many ways appear to mimic or exceed human intelligence, the contours of the capability landscape may diverge fundamentally from expected patterns in human cognition. On the flip side, our findings advise caution when studying generative models for insights into human intelligence and cognition, as seemingly expert human-like outputs may belie non-human-like mechanisms. Overall, the generative AI paradox encourages studying models as an intriguing counterpoint to human intelligence, rather than as a parallel.

## 2    THE GENERATIVE AI PARADOX

We begin by outlining the Generative AI Paradox and an experimental design to test it.

## 2.1 OPERATIONAL DEFINITIONS

Figure 1 offers examples of the seemingly paradoxical behavior of generative models. In language (column B), GPT4 is able to generate a compelling story about 3 friends building a house, but when pressed on details of its *own generated story*, fails to correctly answer a simple question: GPT4 asserts that only one design was sketched in the story despite writing about years of "sketching designs". In vision (column C), a generator produces a correct image beyond average human capabilities, yet the understanding model is unable to single out that correct generation against plausible alternatives, despite selection being the seemingly "easier" task. In both cases, models meet or exceed human generation abilities but lag in understanding.

Observations such as these motivate the Generative AI Paradox:

> *Generative models seem to acquire generation abilities more effectively than understanding, in contrast to human intelligence where generation is usually harder.*

Testing this hypothesis requires an operational definition of each aspect of the paradox. First, what it means for generation to be "more effective" than understanding for a given generative model $m_g$, understanding model $m_u$ and task $t$, with human intelligence as a baseline. Taking **g** and **u** to be some *performance measures* of generation and understanding, we formally state the Generative AI Paradox hypothesis as:

$$\mathbf{g}(\text{human}, t) = \mathbf{g}(m_g, t) \implies \mathbf{u}(\text{human}, t) - \mathbf{u}(m_u, t) > \epsilon \tag{1}$$

Put simply, the hypothesis holds for a task $t$ if a human who achieves the same generation performance **g** as a model $m_g$ would be expected to achieve significantly ($> \epsilon$ for a reasonably large $\epsilon$) higher understanding performance **u** than a model $m_u$ does[2]. In simpler terms, models perform worse on understanding than we would expect of humans with similarly strong generative capabilities. In the language domain,

Generation is straightforward to operationally define: given a task input (question/prompt), generation is the production of observable content to satisfy that input. Thus, performance **g** can be evaluated automatically or by humans (e.g. style, correctness, preference). While understanding is not defined by some observable output, it can be tested by explicitly defining its effects.Thus, we measure performance **u** by asking the following questions:

1. **Selective evaluation.** For a given task, which can be responded to generatively, to what extent can models also select accurate answers among a provided candidate set in a discriminative version of that same task? A common example of this is multiple choice question answering, which is one of the most common ways to examine both human understanding and natural language understanding in language models (Wang et al., 2019) (Figure 1, columns A, C). This tests the performance aspect of understanding, i.e. the ability to identify the answer to a human input.

2. **Interrogative evaluation.** For a given generated model output, to what extent can models accurately respond to questions about the content and appropriateness of that output? This is akin to an oral examination in education (Sabin et al., 2021). (Figure 1, columns B, D ) This tests the explainability aspect of understanding, i.e. the ability to comprehend one's own answer.

These definitions of understanding provide us with a blueprint for evaluating the Generative AI Paradox, allowing us to test whether Hypothesis 1 holds across modalities, tasks, and models.

## 2.2 EXPERIMENTAL OVERVIEW

Here, we provide a high-level road map for experiments informed by the definitions above. We propose 2 sub-hypotheses to test across experimental settings, and provide cross-experiment details.

### 2.2.1 HYPOTHESES

Evaluating whether Hypothesis 1 holds for a given task requires establishing a human baseline, specifically, the understanding performance we expect from a human with the same generation ca-

---

[2]To clarify, the paradox hypothesis is not restricted to the use of a single model to assess both generative and understanding capabilitites; different models can be employed to test these two aspects independently.

pabilities as the model. We define how such a baseline is established for both kinds of understanding above, resulting in 2 sub-hypotheses.

**Selective evaluation.**  Here, we explicitly measure human generation and understanding performance to establish a baseline. We say Hypothesis 1 holds if models underperform in understanding compared to humans with equivalent generation performance (or lower generation performance, assuming that if humans *matched* model generation they would do even better at understanding. The sub-hypothesis is simply:

sub-hypothesis 1: *models meet or exceed humans at generation while lagging at discrimination.*

**Interrogative evaluation.**  For the human baseline here, we assume that humans *can answer simple questions of understanding about their own generations.* For a given task input, we test how accurate models are at answering questions on AI generated outputs and as the human baseline, assume near-perfect accuracy on such questions for their own generations. The sub-hypothesis in this case is:

sub-hypothesis 2: *models struggle to answer simple questions about generated content, which humans could answer for their own generations.*

### 2.2.2  MODELS AND EXPERIMENTS

We focus our study on the strongest current generative models, i.e., those driving interest and concern among experts and the public. We investigate language and vision, modalities where recent impressive progress has been made. We test language models for both generative and understanding capabilities given strong performance in both areas, i.e. taking $m_u = m_g$. We test GPT4 (`gpt-4`) and GPT3.5 (`GPT3.5-turbo`) in a zero-shot setting where we instruct models to output a response given some background information (§A). In contrast, for vision, image generators show weaker understanding (Li et al., 2023a) than dedicated understanding models, and so we assume $m_u \neq m_g$ for vision. We use Midjourney (Inc., 2023) to generate, CLIP (Radford et al., 2021) and OpenCLIP (Ilharco et al., 2021) as understanding models for selective evaluation, and BLIP-2 (Li et al., 2023b), BingChat (Microsoft, 2023), and Bard (Google, 2023) for interrogative evaluation. All results on vision models are obtained in zero-shot fashion.

We conduct experiments across both sub-hypotheses, investigating tasks with selective evaluation of understanding (sub-hypothesis 1) in §3 and investigating tasks with interrogative evaluation of understanding (sub-hypothesis 2) in §4. Both sections include both language and vision tasks.

## 3  CAN MODELS DISCRIMINATE WHEN THEY CAN GENERATE?

First, in our *selective* evaluation, we conduct a side-by-side performance analysis on generative and discriminative variants of tasks to assess models' generation and understanding capabilities in language and vision modalities. We compare this generative and discriminative performance to that of humans. For our tasks we draw on diverse source benchmarks, detailed below:

**Language benchmarks.**  For *dialogue,* we explore two open-ended datasets—**Mutual**$^+$ (Cui et al., 2020) and **DREAM** (Sun et al., 2019), and a document-grounded benchmark, **Faithdial** (Dziri et al., 2022). These tasks require generating coherent continuations based on conversation history (faithful to the document in grounded dialogue). For *reading comprehension*, we include **Topioca** (Adlakha et al. 2022; conversational QA) and **RACE** (Lai et al. 2017; factual QA). For *summarization*, we consider **XSUM** (Narayan et al., 2018). We also include the *commonsense* benchmarks **CommonSenseQA** (Talmor et al., 2019), **SocialIQA** (Sap et al., 2019), **HellaSwag** (Zellers et al., 2019), **PIQA** (Seo et al., 2018), and $\alpha$**NLG/**$\alpha$**NLI** (Bhagavatula et al., 2020). Lastly, we consider the *natural language inference* tasks **WaNLI** (Liu et al., 2022) and $\delta$**-NLI** (Rudinger et al., 2020).

**Vision benchmarks.**  For image generation, we source text prompts from four benchmarks: these range from descriptions of natural scenes, (likely in-domain for the model) to out-of-distribution scenes with specific attributes and relationships that rarely exist in real images. Prompts are sourced from: **COCO** (Lin et al., 2014), **PaintSkill** (Cho et al., 2022), **DrawBench** (Saharia et al., 2022) and **T2ICompBench** (Huang et al., 2023). More dataset details are in §A.2.

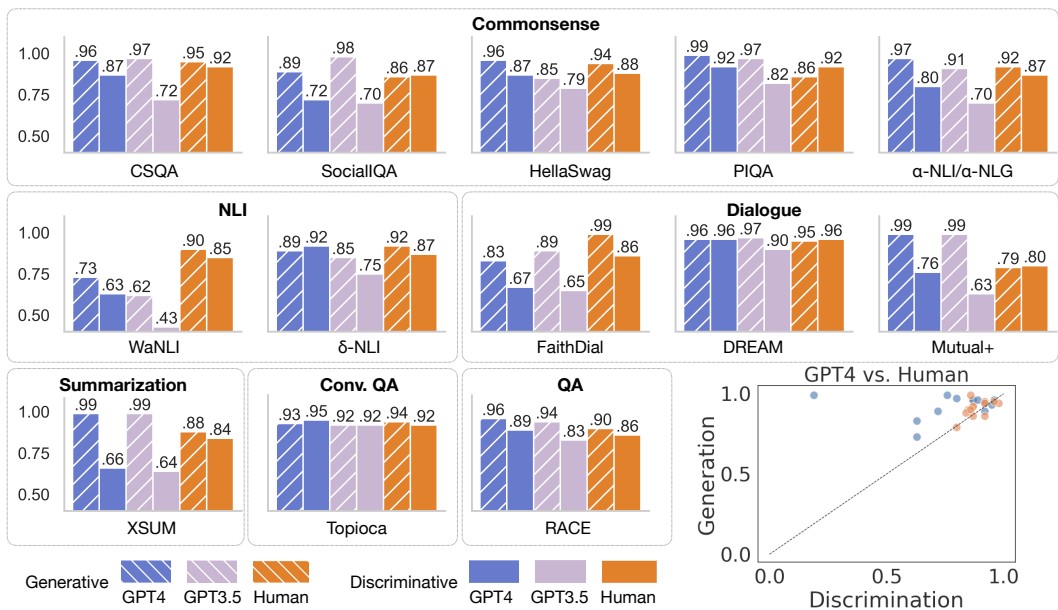

Figure 2: Discriminative and generative performance of GPT3.5 and GPT4 vs Humans. Models outperform humans in generation but underperform them in discrimination for most of the cases. The scatter plot in the bottom right summarizes GPT4's performance vs. human performance (using the hard negatives from Section 3.2 to measure discriminative accuracy for XSUM and FaithDial); each point represents a different task. Humans have a larger positive slope between their discrimination and generation abilities compared to GPT4.

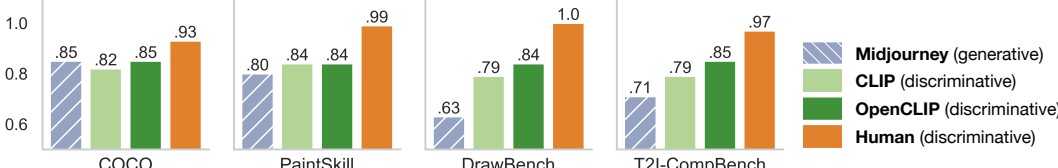

Figure 3: Model and human performance under the generative and discriminative settings on the **vision** modality. We observe models fall short of human accuracy in discriminative performance, and their generative accuracy also lags behind their discriminative accuracy.

**Experimental setup.**    For each task and modality, we consider two settings: **i) generative:** we prompt models to generate a response given task-specific inputs (e.g., dialogue history, document, image caption), and **ii) discriminative:** we require task-specific models to select the correct answer from a set of candidates, using existing candidates where available and otherwise generating options.

For the generative setting, we conduct human evaluations using Amazon Mechanical Turk (AMT) to judge the correctness of the generated responses (i.e, text or image) and report percentage of successful responses satisfying task requirements. For example, for the language domain, we present humans with examples from the language benchmarks.

For the discriminative setting, we report the accuracy of choosing the ground-truth response among the candidate options. To establish a human performance baseline, we ask workers to perform all discriminative tasks and evaluate the correctness of the ground-truth responses for each task.[3] Details of AMT annotations and instructions are in §D.

### 3.1   GENERATIVE AND DISCRIMINATIVE CAPABILITIES IN MODELS VS. HUMANS

**Language.**    Figure 2 presents a comparison of GPT3.5, GPT4, and human generative and discriminative performances. We see that for 10 of the 13 datasets, Sub-hypothesis 1 is supported in at

---

[3]Ground-truth responses were initially written by humans for the language tasks, while ground-truth images are generated by Midjourney.

Figure 4: Model vs. human performance across varying levels of answer difficulty on discriminative tasks.

least one model, with models outperforming humans in generation but underperforming humans in discrimination. For 7 of the 13 datasets, this sub-hypothesis is supported in both models.

**Vision.** It is not practical to ask humans to produce detailed images as we do with vision models, but we assume that an average human could not achieve the stylistic quality of models like Midjourney and thus assume human generation performance is lower. Therefore, we only compare models' generative and discriminative accuracy to humans' discriminative accuracy. Similar to the language domain, Figure 3 shows that CLIP and OpenCLIP[4] fall short of human accuracy in discriminative performance. Assuming human generation is worse, this agrees with sub-hypothesis 1: Vision AI exceeds average humans at generation but lags at understanding.

## 3.2 MODELS FALL FURTHER SHORT OF HUMAN PERFORMANCE WITH HARDER DISCRIMINATION TASKS

We take a closer look at the gap in discriminative performance between humans and models by manipulating the difficulty of the negative candidates. Two types of negatives are considered: **i) Hard negatives**: challenging examples that deter models from relying on data biases and artifacts to produce an answer. These negatives are wrong in subtle and challenging ways; recognizing them may require profound understanding of the task. **ii) Easy negatives**: these candidates are semantically distant from the topic of the question, providing a clear contrast to the correct answer.[5]

Figure 4 (left) shows the comparison between GPT4 and humans[6]. Notably, as the complexity of the candidate answers increases, model performance gradually declines. For instance, in the XSUM task, GPT4 achieves 100% accuracy when selecting the correct answer from easy negatives, but this drops to 19% when confronted with hard negatives. XSUM exhibits a substantial difference in performance compared to FaithDial. Upon inspection, we observe that models tend to make the most mistakes in discrimination tasks when the responses are lengthy and challenging, such as summarizing lengthy documents. In contrast, humans can maintain a consistently high level of accuracy across different levels of difficulty.

Figure 4 (right) shows the discriminative performance of OpenCLIP, in comparison to humans, across difficulty levels. Consistent with the language results, and even more robustly across tasks, we see that while humans show versatile performance across hard and easy negative settings, model performance drops substantially when confronted with hard negatives (from 100% to ∼69%). Overall, these results highlight that humans have the ability to discern correct answers even when faced with challenging or adversarial examples, but we see that this capability is not as robust in LMs. This discrepancy raises questions about the true extent of these models' understanding.

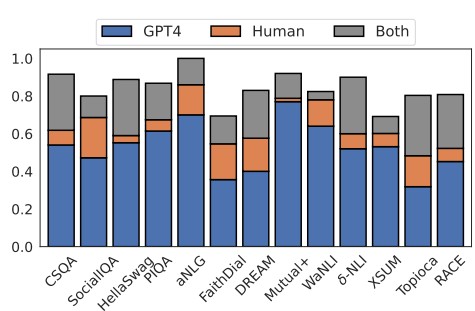

Figure 5: Human's preference scores between human-generated vs. GPT4-generated responses

---

[4]We report the best results on CLIP (`clip-vit-large-patch14`) and OpenCLIP (`CLIP-ViT-bigG-14-laion2B-39B-b160k`), more results can be found in §B.3.

[5]See §B.2 for details about the negative candidates construction. For the language domain, hard negatives are constructed only for tasks that are originally generative in nature (i.e., FaithDial and XSUM).

[6]The same trend also applies for GPT3.5.

## 3.3 MODEL GENERATIONS ARE PREFERRED OVER HUMAN GENERATIONS

To better understand the gap between humans and language models, we asked AMT workers to provide their preferences between machine and human-generated answers in the language-related tasks, along with a rationale for their choices[7]. While both sets of responses score high in correctness (Figure 2), Figure 5 shows a notable trend: workers often favor responses from GPT4 over those generated by humans. The same applies for GPT3.5 (Figure 11 in §B.3). The rationales provided by humans often indicate a preference for GPT4 due to longer response length, more elegant writing style, and being more informative, while human choice is preferred for brevity and conciseness (Figure 12 in §C). This makes the divergence in capabilities–with models excelling in relative terms at generation and humans at understanding-based tasks–even more apparent.

## 4 CAN MODELS UNDERSTAND WHAT MODELS GENERATE?

In the previous section, we showed that models often excel at generating accurate answers while lagging behind humans in the discriminative task. Now, in our *interrogative* evaluation, we investigate to what extent models can demonstrate meaningful understanding of generations—something humans are highly capable of—by directly asking models questions about generated content.

**Language experimental setup.** In language, we first prompt models to generate a paragraph using task-specific background information. Then using its generation as context, we ask the model multiple-choice questions about its own generated information.[8] For example, for **XSUM** (Narayan et al., 2018) (summarization) we prompt the model to generate an article based on a ground-truth summary, and then ask the model to select the best summary (same choices as §3) for the generated article. For **Mutual$^+$** (Cui et al., 2020) (dialogue), the model generates the conversation history that leads to a given dialogue, and then is asked to choose the best dialogue continuing that history. In **HellaSwag** (Zellers et al., 2019) (commonsense), the model generates the context preceding a given sentence and then selects the most fitting continuation for that generated context. We only perform selective evaluation on the *correct generations* verified by humans.

We use zero-shot GPT3.5 and GPT4 for all of the evaluations, both generating and question answering. We report the model generation performance, the selection performance based on content generated by the model, and human selection performance using the model's generated content. As an implicit baseline, we assume that humans can answer such questions about their own generations with high accuracy, and so refrain from the complex process of eliciting these human generations.

**Vision experimental setup.** We conduct interrogative evaluation on image understanding models via visual question answering in an open-ended setting. We consider **TIFAv1.0** (Hu et al., 2023) as the evaluation benchmark, with text prompts from **COCO**, **PaintSkill**, **DrawBench** and **Parti** (Yu et al., 2022). TIFAv1.0 includes questions automatically generated by a language model, only concerning the content specified in the text prompt (*e.g.*, about existence/attributes of an object and relative position between objects). We first ask Midjourney to generate images, based on the text prompts. Then, we interrogate the understanding models (*e.g.*, BLIP-2) with answerable questions (verified by AMT workers) about the generated images. AMT is used to collect human responses, and judge the correctness of human/model outputs. See §C.1 for more details.

**Results.** Results for the language modality are shown in Figure 6 (left). We observe that while the models excel at generation, they make frequent errors in answering questions about their own generations, indicating failures in understanding. Humans, who we assume could not generate such text at the same speed or scale, consistently achieve higher accuracy in QA compared to the model, despite the fact that questions are about the model's own output. As stated in sub-hypothesis 2, we expect humans would achieve even higher accuracy for their own generations. We note that the humans in this study are not experts; producing text as sophisticated as the model's output could be a significant challenge. We anticipate that the performance gap in understanding one's own generation would widen even more when comparing the model to human experts, who are likely to answer such questions with near-perfect accuracy.

---

[7]See Figure 12 in § B.3 for details.

[8]Unlike §3, questions here are about the generation, rather than taking the generation as a potential answer.

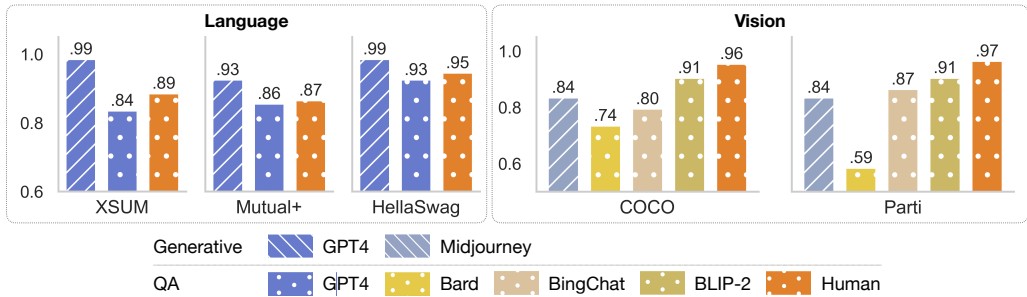

Figure 6: Models vs. human performance on language/visual QA based on model generated texts/images.

Figure 6 (right) shows the interrogative results in the visual modality.[9] We see that image under-standing models still fall short of human accuracy in answering simple questions about elements in the generated images. At the same time, state-of-the-art image generation models can generate images at a quality and speed beyond most average humans (who we expect will have trouble gen-erating comparable realistic images), indicating a relative gap between generation (stronger) and understanding (weaker) in vision AI compared to humans. Surprisingly, the performance gap be-tween models and humans is smaller for simpler models than advanced multimodal LLMs (*i.e.*, Bard and BingChat), which have some intriguing visual understanding abilities, but still struggle to answer simple questions about generated images.

## 5   DISCUSSION

**Assessing the generative AI paradox.**   Broadly, we find significant experimental evidence of the Generative AI Paradox: though models can regularly outperform humans in text and image genera-tion, they fall short of human performance in discriminative versions of generative tasks, and when answering questions about generated content. Furthermore, our analyses show that discrimination performance is more tightly linked to generation performance in humans than in GPT4, and that human discrimination performance is also more robust to challenging inputs. These trends vary across tasks and modalities, but in general our results robustly support the hypothesis that generative capability can outstrip understanding capability in models, especially compared with humans.

**Proposed explanations and points of future study.**   Given the above evidence in support of the Generative AI Paradox, the next question is: *what factors could lead to models that excel at gen-eration even when they cannot demonstrate strong understanding?* We propose some hypotheses below, and encourage future work to explore this question.

Generative AI is defined by the generative learning objective, explicitly encouraging reconstruc-tion/generation of the training distribution, while only implicitly encouraging understanding if it furthers this goal. Human learning, while not completely understood, likely diverges from this by encouraging behavior beyond pure reconstruction of stimuli.

Although we often query generative models as if they were individuals, they typically model a *medium* (e.g. text over many authors in language models). Providing context may push models closer to emulating a specific individual (Andreas, 2022), but they tend towards behavior that looks *distributionally correct* rather than *individually correct*, prioritizing stylistic and document-wide features over details necessary for understanding tasks. Training on many documents (e.g. huge swaths of internet text) also contrasts with humans: it would take an average human reader e.g. over 32 years just to read all the pages of Wikipedia (contributors; Brysbaert, 2019). This obvious discrepancy in not only quantity, but also diversity of knowledge could encourage models to use existing solutions to problems, which they have seen already, whereas humans have not and therefore need to exercise understanding and reasoning to answer the same questions correctly.

Evolutionary and economic pressures can affect the way that AI develops. For instance, popular language model architectures have shown a preference for languages like English (Ravfogel et al., 2019) which has seen the most attention in NLP (Bender, 2019) and thus the most reward for im-

---

[9]We report performance of BingChat, Bard and the best BLIP-2 model (`BLIP2-flan-t5-xxl`) on two subsets, more results can be found in §C.2

provement. Similar pressures could encourage architectures, training paradigms, and other decisions that favor generation over understanding, as generation is harder for humans and thus more useful/valuable. Designing systems that are not affected by the Generative AI Paradox will require understanding its cause. Given the potential explanations above, promising paths forward may involve alternative optimization objectives, limiting the memorization in models to force reasoning, and even incentivizing stronger understanding at a field level.

**Limitations.** Dataset/benchmark contamination is a potential limitation with proprietary models, but this should have similar effects on generation *and* discriminative evaluation in §3, and our evaluation in §4 uses novel generations which would not be seen at training time. Also, we focus on a small set of the most popular/widely used models. Future work should investigate a wider range of models, including smaller or weaker models, for which we hypothesize the paradox may be even more pronounced as we often saw with GPT3.5 vs GPT4 (§3).

While our evaluation of human performance is focused, future work can explore more extensive comparisons between model and human performance. We also advocate for adopting comparison to humans as a widespread practice, to carefully judge when model capabilities extrapolate with human capabilities, and when they do not. Finally, we only investigate *one* divergence between humans and models. Proposing and testing other points of divergence between artificial and natural intelligence exceeds our scope but will be imperative to calm concerns and calibrate excitement.

## 6 RELATED WORK

**Generative paradoxes in large language model behavior.** Prior work paradoxically employs large language models to *improve their own generations*, finding that models successfully identify mistakes (despite these mistakes being generated by the models themselves). Madaan et al. (2023) prompt models to critique and improve their own generations. Agrawal et al. (2023) find that models can identify hallucinated content in their own generations, and Gero et al. (2023) show that models can identify erroneously omitted elements in generated in clinical extraction data.

**Inconsistencies in large language models.** Past work suggests that large language models (LMs) lack a robust concept representation. Dziri et al. (2023) show that strong models often struggle at solving basic tasks like multiplication. Elazar et al. (2021) and Ravichander et al. (2020) show that LMs make inconsistent predictions when prompted with similar statements. Ribeiro et al. (2019) find that QA systems often generate contradictory answers. Kassner & Schütze (2020) and Ettinger (2020) find that models can generate correct facts but also their negations. Jang et al. (2022) construct a benchmark showing large LMs often make inconsistent predictions. Berglund et al. (2023) demonstrate that while models can correctly recognize factual knowledge present in their training data, they fail to make inferences related to those facts.

**Generative models and human cognitive mechanisms.** While the reasoning mechanism of models is unknown, prior work has investigated if models possess similar competencies with humans. Stojnić et al. (2023) evaluate commonsense psychology, finding that while infants can reason about the causes of actions by an agent, models are not capable cannot emulating this. Sap et al. (2022) find that language models fail to demonstrate Theory-of-Mind. Storks et al. (2021) and Bisk et al. (2020) show discrepancies between human and model capacities in physical commonsense reasoning.

## 7 CONCLUSIONS

In this work, we propose the Generative AI Paradox hypothesis, which posits that impressive generation abilities in generative models, by contrast to humans, may not be contingent upon commensurate understanding capabilities. We test this through controlled experiments in language and vision modalities, and though our results show variation depending on task and modality, we find robust support for this hypothesis. Our findings have a number of broader implications. In particular, they imply that existing conceptualizations of intelligence, as derived from experience with humans, may not be applicable to artificial intelligence—although AI capabilities may resemble human intelligence, the capability landscape may diverge in fundamental ways from expected patterns based on humans. Overall, the generative AI paradox suggests that the study of models may serve as an intriguing counterpoint to human intelligence, rather than a parallel.

ACKNOWLEDGEMENTS

This work was funded in part by NSF (DMS-2134012), DARPA MCS program through NIWC Pacific (N66001-19-2-4031), Darpa SemaFor, and the Allen Institute for AI. We thank OpenAI for offering access to various models through the API.

REPRODUCIBILITY

We include a simple description of overall details in §2, as well as experiment-specific details like datasets used and evaluation setup at the beginning of each experiment section, §3 and §C. These descriptions are relatively brief, and we include more extensive information in the appendix. For instance, we include more detail on models, model settings, and datasets in §A. We also include more experimental details and further experiments that can be useful for work comparing to and reproducing our results in §B and §C. Finally, we include more extensive information about our human evaluation templates in §D. All datasets and models we use here are public or can be accessed through public interfaces.

ETHICS STATEMENT

Our work is conducted using existing benchmarks and models, and does not introduce new data, methodology, or models with significant risk of harm. All experiments we conduct would be considered analysis of existing resources, particularly in terms of the performance of models. We conduct human studies, with appropriate IRB exemptions. Based on our estimates of the time for task completion, we ensure workers are paid at least $15 USD per hour. We strive to not conduct any experiments that introduce additional bias, harm, or reduction in diversity, either through the way our research is conducted or its effects. We acknowledge that our work is primarily concerned with certain aspects of performance and does not specifically measure concepts such as bias or toxicity.

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

## A   MODELS AND DATASETS

### A.1   MODELS

For the language domain, We evaluate the performance of 2 LLMs: GPT4 (`gpt-4`) (OpenAI, 2023) and GPT3.5 (`GPT3.5-turbo`) (OpenAI, 2022). The evaluations were conducted from July 2023 to September 2023 using the OpenAI API. During inference, we set nucleus sampling $p$ to 1 and temperature to 1. For each task, we evaluate the performance of each model on 500 test examples.

For the vision domain, we choose the strongest model available to us (*i.e.*, Midjourney (Inc., 2023)) as the image generator. In practice, Midjourney generates 4 images for each text prompt. For image understanding, we evaluate a wide spectrum of models, including variations of CLIP (Radford et al., 2021), OpenClip (Ilharco et al., 2021) for selective evaluation, and BLIP (Li et al., 2022), BLIP-2 (Li et al., 2023b), Instruct-BLIP (Dai et al., 2023), Bard (Google, 2023) and BingChat (Google, 2023) for interrogative evaluation. For all open-source models, we adopt the implementation and model weights available on HuggingFace (Wolf et al., 2019).

---

**Given the dialogue history and the knowledge snippet, kindly generate a response that is faithful to the provided knowledge. In this context, "faithful" implies that every piece of information in the response can be verified as true based on the given knowledge.**

**Knowledge:** Waves suitable for surfing are primarily found in the ocean, but can also be found in lakes or in rivers in the form of a standing wave or tidal bore.

**Dialogue History:**
Speaker A: Sorry to hear you're terrified from sharks. However, did you know that a surfer is someone who can ride on either the forward or deep face of a wave which typically sends her/him close to the shore?
Speaker B: Yeah, what scares me is the sharks, is there any non-ocean waves out there?

**Response:** Speaker A:

---

Figure 7: Example prompt for the FaithDial benchmark used for the zero-shot generative setting.

### A.2   DATASETS

**Language.**   We examined tasks across five categories: commonsense, NLI, dialogue, summarization, and reading comprehension. For tasks inherently discriminative, where the model chooses from a predetermined list of candidates, we omit the construction of negative examples. In generative tasks, we create a list of candidates that includes the groundtruth answer. Specifically, only FaithDial Dziri et al. (2022) and XSUM Narayan et al. (2018) fall under the generative category, while the remaining benchmarks are designed in a discriminative manner. Refer to §B.2 for details on negative candidate constructions.

Here, we delve into how we evaluate the dialogue tasks, noting that the same procedure applies to the rest of the tasks. Within the dialogue tasks, our focus spans three benchmarks: DREAM Sun et al. (2019), Mutual+ Cui et al. (2020), where the objective is to generate a coherent continuation based on the conversation history, and FaithDial Dziri et al. (2022), where the goal is to produce a faithful response using both the document and the conversation history.

For the **generative evaluation** in the dialogue task, we input the model with the dialogue history, and if applicable, the knowledge snippet. We instruct the model to generate a response that is coherent or faithful, depending on the provided knowledge. For the **discriminative evaluation** for the same objective, we prompt the model to select the correct response from a list of answers, considering the

> **Given the dialogue history and the knowledge snippet, please select the response from the options below that is faithful to the provided knowledge. In this context, "faithful" indicates that every piece of information in the response can be verified as true based on the given knowledge.**
>
>
> **Knowledge:** Waves suitable for surfing are primarily found in the ocean, but can also be found in lakes or in rivers in the form of a standing wave or tidal bore.
>
> **Dialogue History:**
> Speaker A: Sorry to hear you're terrified from sharks. However, did you know that a surfer is someone who can ride on either the forward or deep face of a wave which typically sends her/him close to the shore?
> Speaker B: Yeah, what scares me is the sharks, is there any non-ocean waves out there?
>
> **Option 1:** For sure there is. An alternative to ocean waves could be lakes and rivers which have a phenomenon called standing waves. Have you ever watched people surf?
> **Option 2:** Sure, keep in mind that it is a type of courting. You and your girl participate in social activities, but you don't have to be alone, as others can be there as well.
> **Option 3:** Oh no that is terrible and I am sorry to hear that.
> **Option 4:** That's cool, do you know who the first person to climb it was?
>
> **Response:**

Figure 8: Example prompt for the FaithDial benchmark used for the zero-shot discriminative setting.

conversation history and knowledge. Please refer to Figure 7 for the prompt in the generative setting and Figure 8 for the prompt in the discriminative setting for the FaithDial benchmark.

**Vision.** For **selective evaluation**, we source text prompts from 4 datasets, COCO (Lin et al., 2014), Paintskill (Cho et al., 2022), DrawBench (Saharia et al., 2022) and T2ICompBench (Huang et al., 2023). COCO prompts are human-written captions on real images. PaintSkill features text prompts that examine image generation on specific object categories, object counts and spatial relations between objects. DrawBench additionally test for long-form text, rare words, and challenging prompts. T2ICompBench is designed to test models on open-world, compositional text-to-image generation, with text prompts covering 3 categories, attribute binding, object relationships, and complex compositions. For **interrogative evaluation**, we consider TIFAv1.0 (Hu et al., 2023) as the evaluation benchmark. The text prompts in TIFAv1.0 are originally from COCO, Paintskill, DrawBench and Parti (Yu et al., 2022). For each text prompt, TIFAv1.0 includes questions automatically generated by a language model, only concerning the content specified in the text prompt (*e.g.*, about existence/attributes of an object and relative position between objects).

## B    CAN MODELS DISCRIMINATE WHEN THEY CAN GENERATE?

### B.1    SETUP

**Language.** The basic setup for this evaluation is shown in Figure 9 (left).

**Vision.** We follow the setup on language tasks and consider two settings on each dataset for evaluation: **i) generative**: we prompt Midjourney to generate images given the text descriptions, and **ii) discriminative**: we require the image understanding models to select the image, that better matches the text description, from two candidates. For the generative setting, we conduct human evaluations

## Selective Evaluation

**Prompt**
Q: What is the capital of Canada?

**Generation**
**Ottawa**

**Understanding**
a. Washington DC
b.    Toronto
c.    Ottawa
d.    London

## Interrogative Evaluation

**Prompt**
Q: Write a story about about 3 characters baking a cake

**Generation**
**Story: Once upon a time, Tim, Sally, and Steve decided …**

**Understanding**
Q: In the story, did Tim add frosting before or after the cake was baked?

A: **Tim added frosting after**

Figure 9: Diagram illustrating the two evaluation settings on language modality, *selective evaluation* (left) and *interrogative evaluation* with the model output in each case highlighted in orange. In selective evaluation, we compare generated responses to selected responses for the same prompt. For interrogative evaluation, we test the ability of models to answer questions about their own generations.

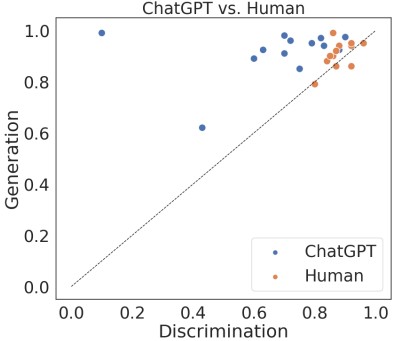

Figure 10: GPT3.5 vs. Humans. Humans show a larger positive correlation between their discrimination and generation abilities compared to GPT3.5.

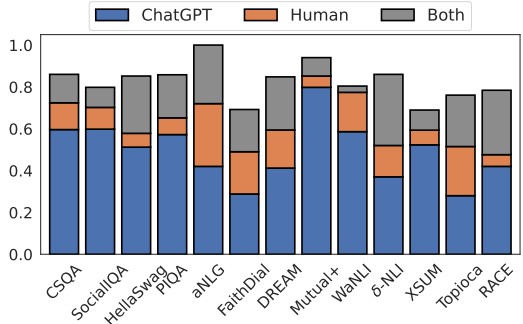

Figure 11: Quality scores of human-generated responses vs. GPT3.5 response scores

on AMT to judge whether the generated image matches the text prompt. In total, we randomly sample 100 text prompts per dataset. As Midjourney generates 4 images for each text prompt and users of Midjourney in practice would pick the best image among the four, we report the success rate per prompt as the evaluation metric. For the discriminative setting, we construct the candidates with a negative image for each positive image from the successful generations verified by human workers of a given prompt. We report accuracy as the evaluation metric. Human performance on discriminative setting is measured by comparing the majority of 3 human responses to a 4th one.

### B.2 NEGATIVE CANDIDATES CONSTRUCTION

**Language.** To construct the negative examples for the FaithDial and XSUM datasets, we explore two corruptions processes:

1. **Easy negatives**: we compile responses that are unrelated to the information provided in the knowledge snippet $K$ (such as a dialogue or summary document). For a given context, we randomly select a gold response that was based on a different $K$.

2. **Hard negatives**: To generate examples that are likely hallucinations but sufficiently challenging to distinguish from correct answers, we directly perturb the knowledge spans $K$ and then feed them to GPT4. We replace up to two entities in the original $K$ with entities of the same type from the same document to avoid easy-to-detect off-topic entities. The re-

sponse generated by GPT4 will be a hallucination, containing subtle alterations that render it incorrect when compared to the groundtruth response. For each task, we consider three negative candidates.

**Vision.** To examine the discriminative performance gap of image understanding models across different difficulty levels, we similarly construct hard and easy negatives in the image space to evaluate image understanding models: **i) Hard negative**: a negative image that is generated based on the same text prompt as the positive image, such that it is semantically close to the text prompt, but contains subtle mistakes identifiable by humans. **ii) Easy negative**: a negative image that is randomly sampled from the successful generations of a different prompt in the same dataset (as the positive image), such that it is semantically distant from the positive image and can be easily distinguishable. For both cases, we use AMT to verify the negative samples and only retain the ones with agreeable judgments among 3 workers. In the end, we have 345 instances with hard negatives, including 52, 72, 100 and 42 instances for COCO, PaintSkill, CompBench and DrawBench, respectively; and 372 instances with easy negatives, comprising 82, 72, 100 and 42 instances for COCO, PaintSkill, CompBench and DrawBench, respectively.

### B.3 ADDITIONAL RESULTS

**Language.** In Figure 10, we show humans exhibit a larger positive correlation between their discrimination and generation abilities compared to GPT3.5. Figure 11 illustrates that workers often favor responses from GPT3.5 over those generated by humans. Figure 12 shows the rationales provided by humans on their preferences for GPT4 responses compared to groundtruth human responses.

**Vision.** We include additional results from different model variants of CLIP and OpenCLIP in Table 1. These models consistently fall short of human accuracy in discriminative performance. In Table 2, we observe the gap between model and human performance becomes larger as the difficulty level of the task increases with easy and hard negatives.

Table 1: Additional results on selective evaluation for vision modality.

|  | COCO | PaintSkill | T2ICompBench | DrawBench |
|---|---|---|---|---|
| Midjourney (Generative) | 85.00% | 80.41% | 71.15% | 62.63% |
| *Discriminative* | | | | |
| Human | **92.86%** | **99.30%** | **97.00%** | **100.00%** |
| CLIP | | | | |
| clip-vit-base-patch16 | 79.81% | 75.00% | 79.00% | 76.19% |
| clip-vit-base-patch32 | 83.66% | 72.39% | 77.50% | 77.38% |
| clip-vit-large-patch14 | 85.58% | 81.95% | 84.50% | 78.57% |
| clip-vit-large-patch14-336 | 87.50% | 78.47% | 81.50% | 76.19% |
| OpenCLIP | | | | |
| CLIP-ViT-bigG-14-laion2B-39B-b160k | 81.73% | 85.28% | 84.50% | 84.53% |
| CLIP-ViT-g-14-laion2B-s12B-b42k | 82.70% | 85.41% | 83.50% | 77.38% |
| CLIP-ViT-g-14-laion2B-s34B-b88K | 83.66% | 81.25% | 88.00% | 79.76% |
| CLIP-ViT-H-14-laion2B-s32B-b79K | 82.69% | 85.41% | 83.50% | 77.38% |

## C  CAN MODELS UNDERSTAND WHAT MODELS GENERATE?

The basic setup for this evaluation is shown in Figure 9 (right).

### C.1  EXPERIMENTAL SETUP.

**Language.** We additionally explore *constrained* generation in which models are given lexical constraints for generation. In the *constrained* setting, we use a compositional task that covers diverse

Table 2: Additional results on model vs. human performance across varying levels of answer difficulty for vision tasks.

| | COCO | | PaintSkill | | T2ICompBench | | DrawBench | |
|---|---|---|---|---|---|---|---|---|
| | Hard | Easy | Hard | Easy | Hard | Easy | Hard | Easy |
| Human | **85.71%** | **100%** | **98.61%** | **100%** | **94.00%** | **100%** | **100%** | **100%** |
| **CLIP** | | | | | | | | |
| clip-vit-base-patch16 | 59.62% | 100% | 50.00% | 100% | 58.00% | 100% | 52.38% | 100% |
| clip-vit-base-patch32 | 67.31% | 100% | 45.83% | 98.95% | 55.00% | 100% | 54.76% | 100% |
| clip-vit-large-patch14 | 71.15% | 100% | 63.89% | 100% | 69.00% | 100% | 57.14% | 100% |
| clip-vit-large-patch14-336 | 75.00% | 100% | 56.94% | 100% | 63.00% | 100% | 52.38% | 100% |
| **OpenCLIP** | | | | | | | | |
| CLIP-ViT-bigG-14-laion2B-39B-b160k | 63.46% | 100% | 70.83% | 99.74% | 69.00% | 100% | 69.05% | 100% |
| CLIP-ViT-g-14-laion2B-s12B-b42k | 65.39% | 100% | 70.83% | 100% | 67.00% | 100% | 54.76% | 100% |
| Clip-ViT-g-14-laion2B-s34B-b88K | 67.31% | 100% | 62.50% | 100% | 76.00% | 100% | 59.52% | 100% |
| CLIP-ViT-H-14-laion2B-s32B-b79K | 65.38% | 100% | 70.83% | 100% | 67.00% | 100% | 54.76% | 100% |

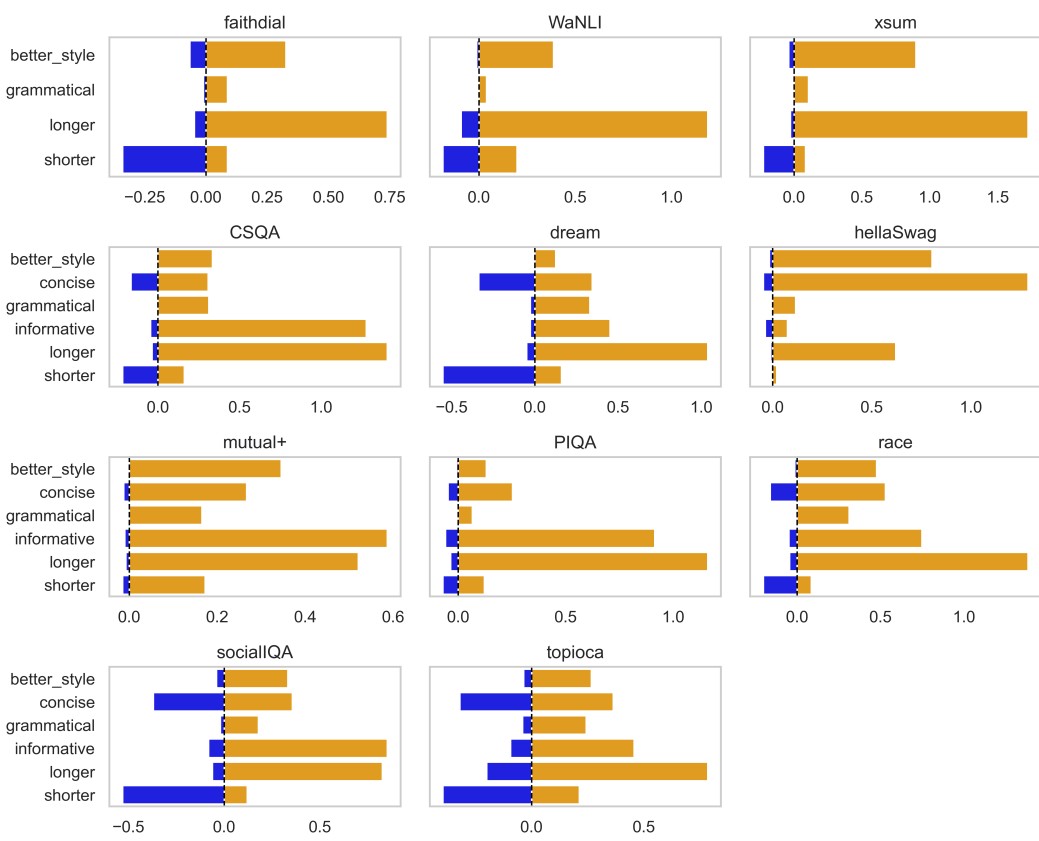

Figure 12: AMT Workers rationals on their preferences for GPT4 responses compared to groundtruth human responses.

Table 3: Example of tasks in Collie Benchmark covering several generation levels including word, sentence, paragraph and passage.

| Task | Example |
|---|---|
| word01 | Generate a word with at least 15 letters. |
| word02 | Generate a word with 10 letters, where letter 1 is 's', letter 3 is 'r', letter 9 is 'e'. |
| word03 | Generate a word with at most 10 letters and ends with 'r'. |
| sent01 | Please generate a sentence with exactly 82 characters. Include whitespace into your character count. |
| sent02 | Generate a sentence with 10 words, where word 3 is "soft" and word 7 is "beach" and word 10 is "math". |
| sent03 | Generate a sentence with at least 20 words, and each word less than six characters. |
| sent04 | Generate a sentence but be sure to include the words "soft", "beach" and "math". |
| para01 | Generate a paragraph where each sentence begins with the word "soft". |
| para02 | Generate a paragraph with at least 4 sentences, but do not use the words "the", "and" or "of". |
| para03 | Generate a paragraph with exactly 4 sentences, each with between 10 and 15 words. |
| para04 | Generate a paragraph with at least 3 sentences, each with at least 15 words. |
| para05 | Generate a paragraph with 2 sentences that end in "math" and "rock" respectively. |
| pass01 | Generate a passage with 2 paragraphs, each ending in "I sit." and "I cry." respectively. |

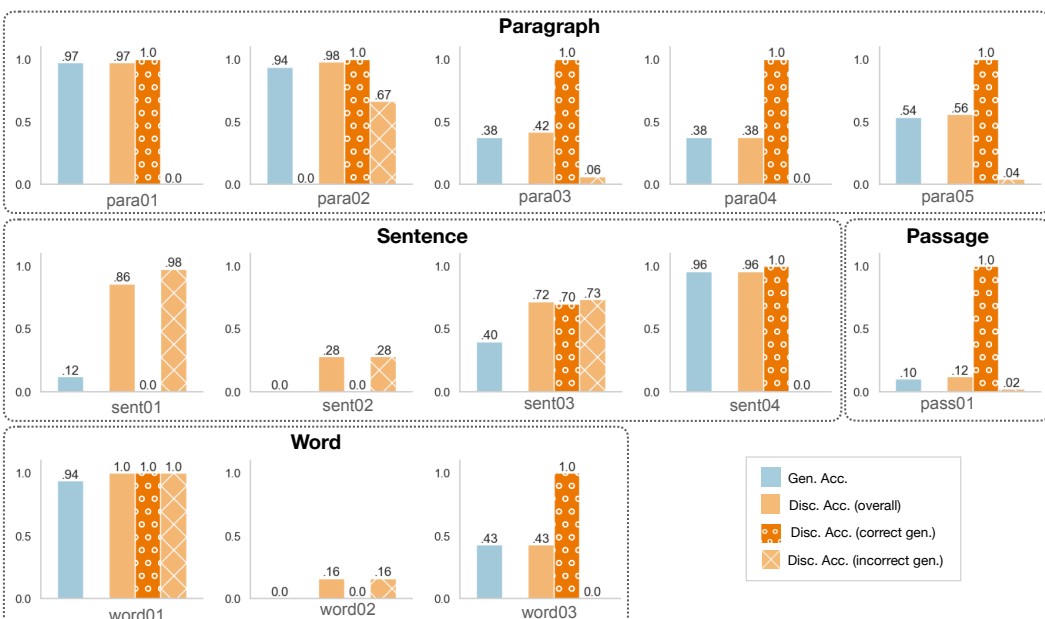

Figure 13: GPT4 Generative Constraint Satisfaction on Collie along with discriminative accuracy on its Generations.

generation levels, i.e., word, sentence, paragraph, and passage: COLLIE-v1 (Yao et al., 2023), which contains 2,080 constraint instances across 13 different task types shown in Appendix Table 3. We generate outputs for 50 examples per task. We then ask models about their generations, specifically querying about whether the generations satisfy the given constraints.

**Vision.** For interrogative evaluation on vision modality, we randomly sample 25 prompts from each subset of TIFAv1.0, resulting in 100 prompts in total. For evaluation of image understanding models, we include all answerable questions on the generated images (verified by AMT workers) from the original dataset, and collect the groundtruth answers on this questions from human annotators. Note that even when the generated image does not strictly align with the text prompt, we still include the image-question pairs that are considered answerable by human annotators to interrogate understanding models. In the end, we gather 1,871 image-question pairs, with 533, 482, 422 and 434 instances on COCO, Paintskill, DrawBench and Parti subset, respectively. Human performance is measured by comparing the majority of 3 human responses and the 4th one.

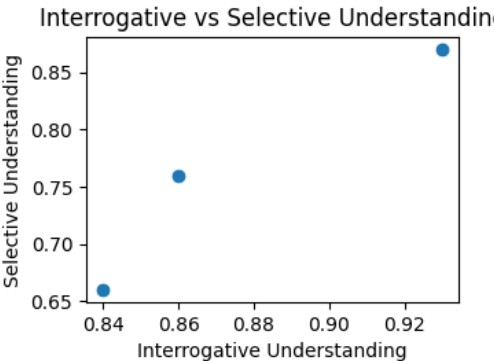

Figure 14: A comparison of interrogative and selective understanding in GPT-4 on the 3 tasks they are both tested for: XSUM, Mutual+, and HellaSwag. They generally seem to correlate, but this will require further study.

## C.2 ADDITIONAL RESULTS

**Language** We report on the constrained setting. Figure 13 shows GPT4's constraint satisfaction rate across 13 tasks in Collie . Certain tasks with simple constraints such as `word01` (generating a word with a minimum number of letters), and `sent04` (generating a sentence containing three specific words) are less challenging for models. However, we observe a significant drop in performance when posed with arbitrary position constraints and strict counting requirements (e.g., `sent02`, generating a sentence with x words, where the 3rd word is A, and the 7th word is B, ...), suggesting that current models cannot handle generations when faced with rich compositional constraints. Unlike the open-ended setting, we find models are often better at answering questions about their generations than generating. We propose that more precise constraints like this are easier to judge (trivial for humans) while being much harder for models to exercise flexible generation over.

Further, we compare performance between interrogative and selective understanding for GPT-4 across the 3 tasks they are both tested on, finding a possible correlation (Figure 14). This lends some support to the overall difficulty of task understanding being related between different notions of understanding, possibly due to general attributes of the task such as complexity of the underlying text, relationship between input and output, as well as the difficulty of the underlying style of the text.

**Vision** Table 4 shows the full results from different model variants of BLIP, BLIP-2, Instruct-BLIP, Bard and BingChat on all 4 subsets of TIFAv1.0. Note that Bard and BingChat can occasionally refuse to answer the question, when the image contains people. The results from these models are on a subset when they can provide a reasonable answer. The model performance is consistently lower than human performance, acorss different models.

## C.3 QUALITATIVE EXAMPLES

Here, we include a small-scale study we conducted on GPT-4, of the model's ability to answer questions about stories it generates. Prompts are constructed by the paper authors, and questions are constructed by hand to allow probing of specific details of the generated content not specifically depending on the prompt. We specifically focus on simple questions that the model nonetheless gets wrong. These examples are in Tables 5, 6, 7.

Moreover, we add more qualitative examples on vision modality in addition to what have been shown in Figure 1. Examples of model outputs for selective setting are shown in Figure 15, and those for interrogative setting are in Figure 16.

Table 4: More results on interrogative evaluation for vision modality.

|  | COCO | PaintSkill | DrawBench | Parti |
|---|---|---|---|---|
| Midjourney (Generative) | 84.00% | 52.00% | 72.00% | 84.00% |
| *Questiong Answering* | | | | |
| Human | **95.88%** | **97.72%** | **96.32%** | **96.83%** |
| `blip-vqa-base` | 89.68% | 83.82% | 82.23% | 84.56% |
| `blip-vqa-capfilt-large` | 89.68% | 83.82% | 82.23% | 84.56% |
| `BLIP2-flan-t5-xl` | 89.49% | 83.20% | 81.52% | 85.25% |
| `BLIP2-flan-t5-xxl` | 91.18% | 88.59% | 85.31% | 90.56% |
| `BLIP2-opt-2.7b` | 81.99% | 82.16% | 72.75% | 79.03% |
| `BLIP2-opt-6.7b` | 84.05% | 77.39% | 72.99% | 75.81% |
| `instructblip-flan-t5-xl` | 88.56% | 81.54% | 81.99% | 88.25% |
| `instructblip-flan-t5-xxl` | 91.93% | 84.02% | 84.83% | 88.02% |
| `instructblip-vicuna-7b` | 92.50% | 83.20% | 81.75% | 87.10% |
| `instructblip-vicuna-13b` | 88.74% | 80.71% | 78.67% | 76.04% |
| Bard | 74.02% | 66.28% | 56.33% | 59.42% |
| BingChat | 80.49% | 87.20% | 80.68% | 87.20% |

## D HUMAN ANNOTATION ON AMT

All human annotations were conducted on the Amazon Mechanical Turk (AMT). Through a paid qualification round, we qualify 130 best performing workers that consistently provide conscientious, high-quality annotations. This project paid the Mturk workers between $15-25 per hour in median pay depending on the difficulty of the task. We report on the pairwise agreement rate [10] : the agreement levels range from 90-97% over the datasets.

**Human Discrimination Evaluation.** For the language modality, we obtain human discrimination numbers by prompting the AMT worker with the appropriate context and question for the given task, and ask them to choose the correct response among a list of choices. For vision modality, the set up is the same with one exception: the workers are asked to choose the best *matching* image for the caption. Each examples were annotated by 3 workers and majority vote was taken for the reported results. An example of discriminative human evaluation is found in Figure 17.

**Generation Evaluation.** For model generation evaluation in the language modality, the worker is given the context, question, and the model generation and is asked to say if the generation is an acceptable response. In the vision modality, the same evaluation is conducted by prompting the worker with a caption (the input prompt) and the generated image and asked if the image matches the caption. Each examples were annotated by 3 workers and majority vote was taken for the reported results. Evaluation of groundtruth also uses this template. Template used is found in Figure 18.

**Comparative Evaluation.** For language modality only, we conduct the comparative evaluation. The worker is prompted with the appropriate context and question, and given model generation and groundtruth answer, asked to choose which is the preferred answer. In this setup, they are also asked to choose from 3-5 checklist that seeks to ascertain the rationale for the choice (e.g., "The response is shorter and so more to the point" or "The style is better"). Template used is found in Figure 19.

**Human Writing.** For vision modality only, we conduct a writing task where the worker is prompted with a generated image and a question, and asked to briefly answer the question. Each examples were annotated by 2 workers: 1 to establish groundtruth answers for generated images based

---

[10]Our data is skewed to a single label. Measures like kappa or Krippendorf's alpha are affected by label imbalance and can lower the score even when the agreement is high Feinstein & Cicchetti (1990). This makes them unsuitable for interannotator reliability measures in such conditions Xie (2013).

Table 5: Qualitative questioning of GPT-4 about its own generated stories

| **Prompt** |
| --- |
| Write a two paragraph story about 3 people building a house |

| **GPT-4** |
| --- |
| Once upon a time, in the heart of the serene countryside, lived three close friends: Bob, an architect; Allen, a handyman; and Chris, a carpenter. They were always dreaming about creating something exceptional and finally, after years of saving money and sketching designs, they decided to build their dream house. Bob, with his architectural expertise, sketched a breathtaking design of a two-storied country house with a wooden porch. Allen collected all the necessary tools, and Chris with his woodworking skills, pledged to make every single piece of furniture unique. |
| Weeks turned into months, and the three friends tirelessly worked day and night to build the house. Bob and Chris constructed the wooden framework, walls, and roof; Allen managed the plumbing and electrical fittings. During their breaks, they would sit together on the raw framework, overlooking the rolling green hills, discussing the interior decoration and envisaging the fulfilment of their dream. Despite the challenges, disagreements, and exhaustion, they never gave up and encouraged one another to keep going. After several months of strenuous efforts, the house was finally built to match their grand blueprint. It was a testament to the trio's diligent work, creativity, and friendship, and they named it 'The Bond House'. |

| **Question** |
| --- |
| How many times did Bob sketch the house? |

| **GPT-4** |
| --- |
| The story mentions that Bob sketched the house once. |

| **Error** |
| --- |
| We know Bob sketched many designs over the years. |

| **Question** |
| --- |
| Was the very first sketch of the house two stories? |

| **GPT-4** |
| --- |
| Yes, the very first sketch of the house was two stories. |

| **Error** |
| --- |
| We know Bob sketched designs before the first one described, so we actually don't know about the very first sketch. |

| **Question** |
| --- |
| Was plumbing added before or after the walls and roof were constructed? |

| **GPT-4** |
| --- |
| The story suggests that the plumbing was added after the walls and roof were constructed. |

| **Error** |
| --- |
| Plumbing is typically added before the walls, as plumbing is inside of the walls. |

| **Question** |
| --- |
| What did they likely do on the last break before the house was done? |

| **GPT-4** |
| --- |
| They likely sat together on the raw framework, overlooking the rolling green hills, discussing the interior decoration and envisaging the fulfilment of their dream. |

| **Error** |
| --- |
| The raw framework was unlikely to be exposed by the time they took their last break. |

on majority vote (expert adjudication was used in case of disagreement) and 1 to gauge human performance on the task. Template used is found in Figure 20.

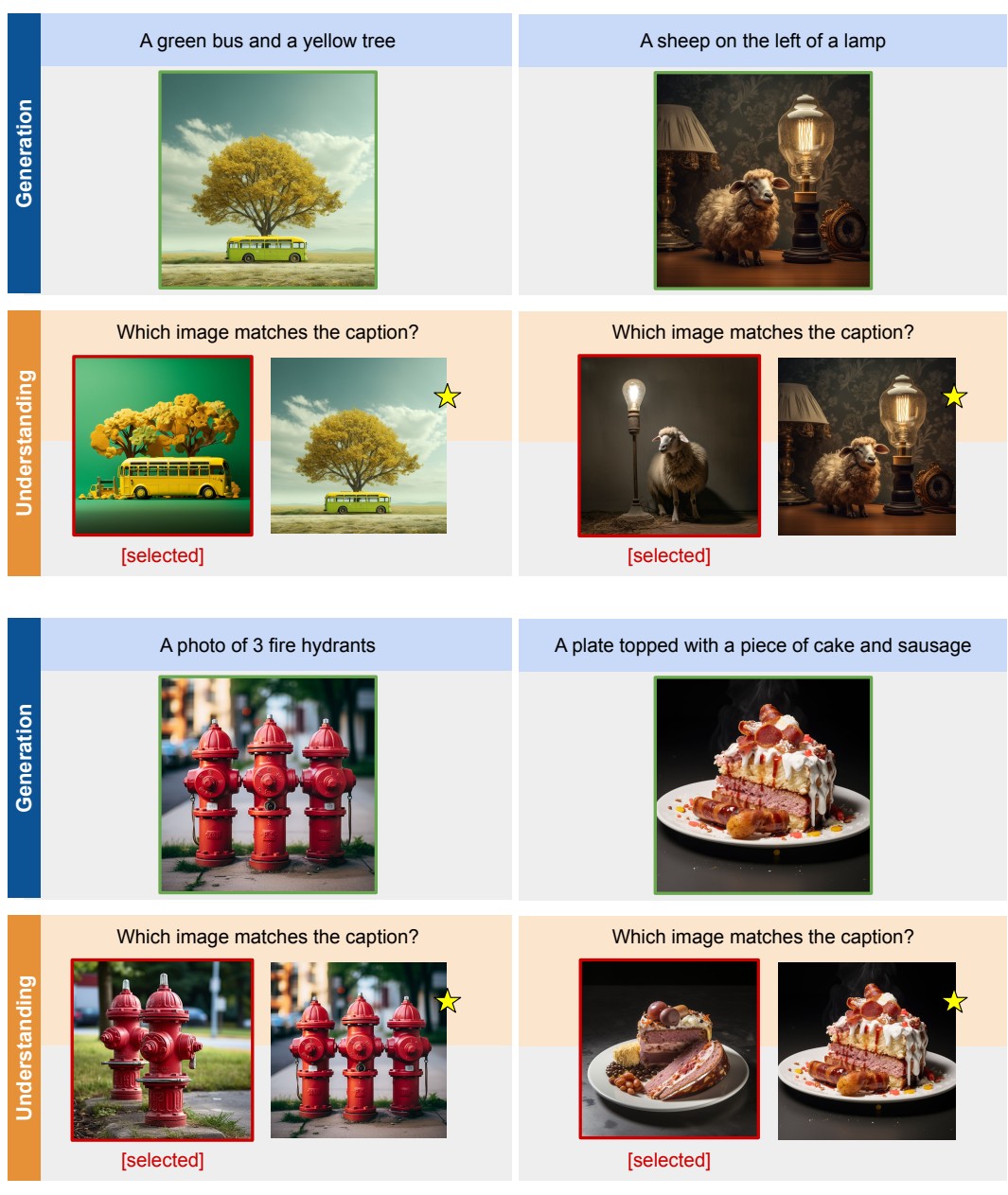

Figure 15: Qualitative examples of selective setting on vision modality.

Table 6: Qualitative questioning of GPT-4 about its own generated stories

**Prompt**
Write a 5 paragraph story about 3 friends finding treasure

**GPT-4**
Once upon a time, in the small, peaceful town of Verdi, three friends- Tom, Max, and Sara lived. Tom was the most adventurous and curious in the group. Max, the tallest among them, was the most athletic and helped Tom in their ventures. Sara, the most sensible, constantly discouraged their dangerous adventures but couldn't resist tagging along. One day, Tom, investigating an old myth about hidden treasure, decided to persuade his friends to join him in his hunt.

The trio cautiously made their way through the dense woods that bordered the town in the direction of the hill as stated in the legend. Thick webs of strong vines and thorny bushes were a challenge, but Max's strength and Sara's whittled down hiking stick bore a path for them. As they reached the clearing at the top of the hill, they saw a decrepit stone structure, obscured by overgrown foliage and moss. The friends couldn't contain their excitement – it was very likely the location of the treasure according to the myth.

Inside the crumbling stone structure, there were ancient carvings and paintings along the dusty walls. Although time-worn, they could still make out the drawings. The walls told the story of a mighty king and his beloved, resplendent treasure. Sara, with her keen eyes, discovered a hidden passage in the corner of the room. The friends entered the passage with bated breath, their flashlight rays cutting through the darkness. A dusty chest lay quietly in the musty air, the very image of the treasure chest Tom had described from the legend. Anticipation crackled in the air as they all held their breath while Tom painstakingly opened the heavy lid.

The sight that greeted them was beyond their wildest dreams. The chest was filled with gold, silver, and precious gems of every color. There were ornate arm bands, necklaces, and coins - glimmering in the torchlight, waiting for centuries to be discovered. They looked at each other, their faces mirroring the shock and ecstasy of their incredible discovery.

The trio didn't let their triumph consume their logic. They understood such a treasure was part of their town's history, and stealing it would be morally wrong. Keeping their excitement at bay, they informed the town authorities about their finding. The town celebrated the discovery, but the greatest joy was in the hearts of Tom, Max, and Sara. While they may not have gained materialistic wealth, their bond strengthened, and they discovered the true treasure of friendship and adventure. This adventure was a tale they would reminisce fondly about, well into their old age. The friends garnered respect and admiration from the townsfolk, but they remained humble, attributing their discovery more to luck than skill. Their friendship had always been cherished, but now it was legendary. The tale of their adventure became a cherished lore, passed down generations, speaking of a time long ago, when three friends discovered a hidden treasure but chose their friendship and integrity over greed.

|  |
|--|

**Question**
Did Sara discourage their adventure to find the treasure?

**GPT-4**
Yes, Sara initially discouraged their adventure to find the treasure.

**Error**
The story does not say this explicitly.

|  |
|--|

**Question**
Who would a reader expect the trio to inform about the treasure once they found it?

**GPT-4**
A reader would expect the trio to inform the town authorities about the treasure once they found it.

**Error**
This is actually a surprising part of the story. Generally, a reader would expect them to keep the treasure.

|  |
|--|

**Question**
What is the most surprising part of the story?

**GPT-4**
The most surprising part of the story is when the friends, after finding the treasure, decide not to keep it for themselves but instead inform the town authorities about their discovery.

**Error**
Although this is a correct answer, it contradicts what the model responded to the previous question.

Table 7: Qualitative questioning of GPT-4 about its own generated stories

**Prompt**
Write a 2 paragraph story about 3 scientists building a space ship

**GPT-4**
Once upon a time in the outskirts of London, three brilliant scientists, Dr. Benjamin, Dr. Amelia, and Dr. Max, dreamt of launching a spaceship to enter the vast, uncharted universe. Their burning ambition and unparalleled intellect dumped them in countless sleepless nights, where they poured over blueprints, mathematical equations, and stacks of astronomical data. The trio envisioned building a state-of-the-art spaceship powered by a revolutionary plasma propulsion technology. Not only would their ship be faster than any mankind had ever built, but it would also have the capability of sustaining life for centuries, hinting at the prospect of interstellar travel.
After years of relentless effort and dedication, their dream finally transformed into a magnificent spaceship named Exodus. It was unlike anything the world had ever seen before. Spanning more than two hundred feet in length, the ship looked like a cosmic shark decked with arrays of communication antennae and numerous high-tech observational instruments. The generous interior boasted a miniature ecosystem, advanced cryogenic facilities, and fully automated systems designed to withstand the extremities of deep space. As the world applauded the triumphant spectacle, the three scientists stood smiling, gazing at their incredible creation and dreaming of the extraordinary journeys it would soon embark upon.

**Question**
Did the trio pour over blueprints, mathematical equations, and stacks of astronomical data before or after deciding to build a state-of-the-art spaceship?

**GPT-4**
The trio poured over blueprints, mathematical equations, and stacks of astronomical data before deciding to build a state-of-the-art spaceship.

**Error**
The story actually seems to imply that they did this *because* they wanted to build a state-of-the-art spaceship, which would imply it happened *after*.

**Question**
Why did the trio pour over blueprints, mathematical equations, and stacks of astronomical data?

**GPT-4**
The trio poured over blueprints, mathematical equations, and stacks of astronomical data because they were planning to build a state-of-the-art spaceship for interstellar travel.

**Error**
While correct, this contradicts the previous answer.

**Question**
Can we say anything about how long the three brilliant scientists knew each other before Exodus was built?

**GPT-4**
The story does not provide information on how long the three brilliant scientists knew each other before Exodus was built.

**Error**
We can say at least "years" because the trio were working together on the ship for years.

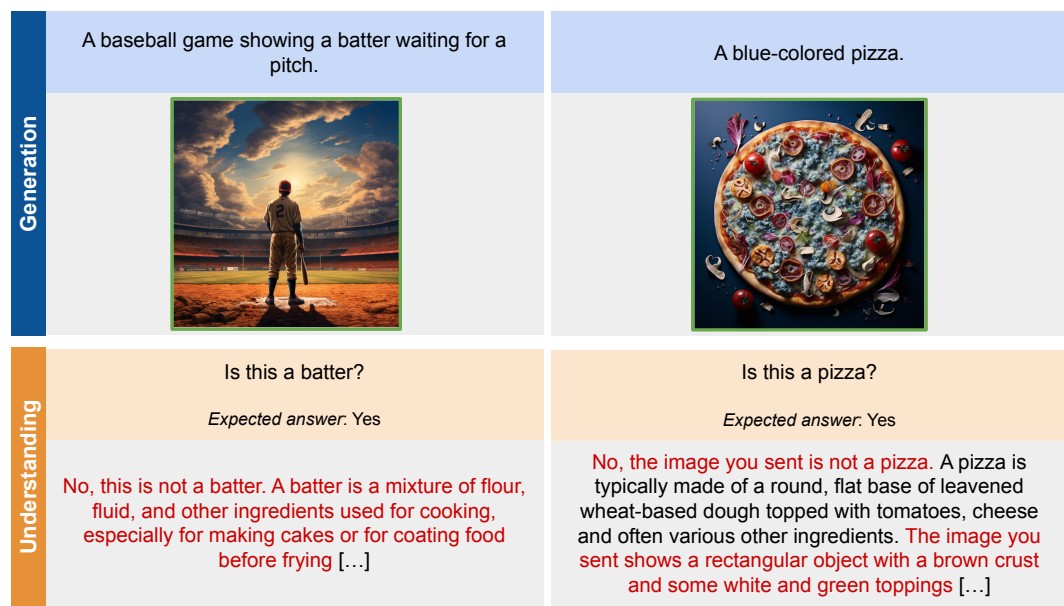

Figure 16: Qualitative examples of interrogative setting on vision modality.

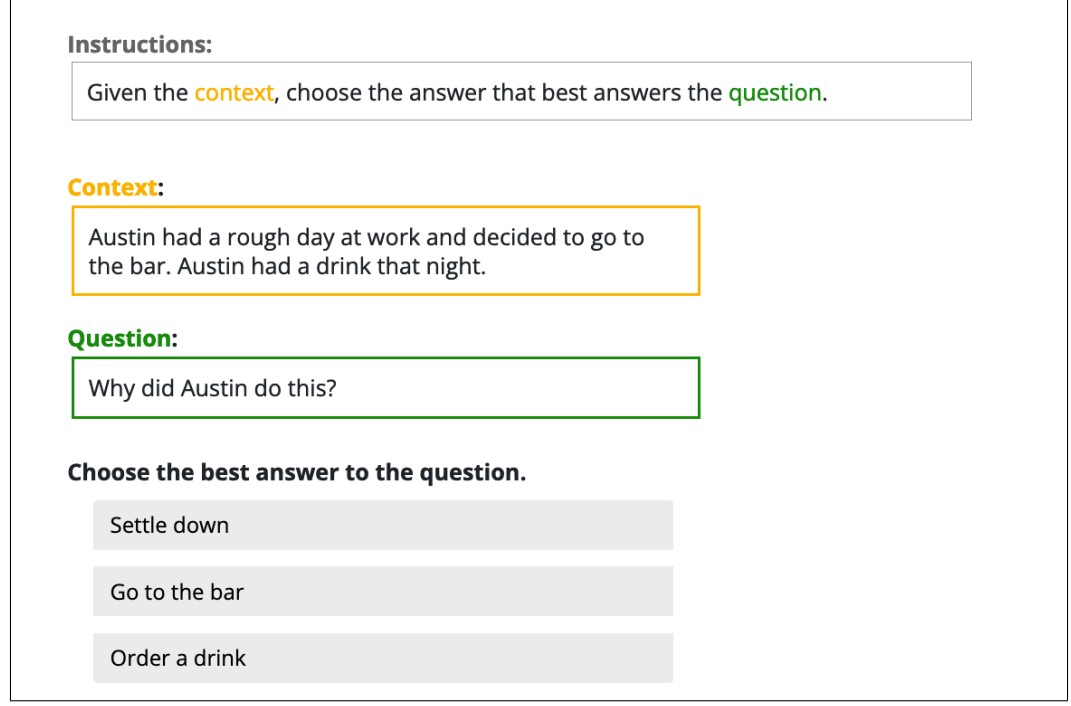

Figure 17: Human discrimination template.

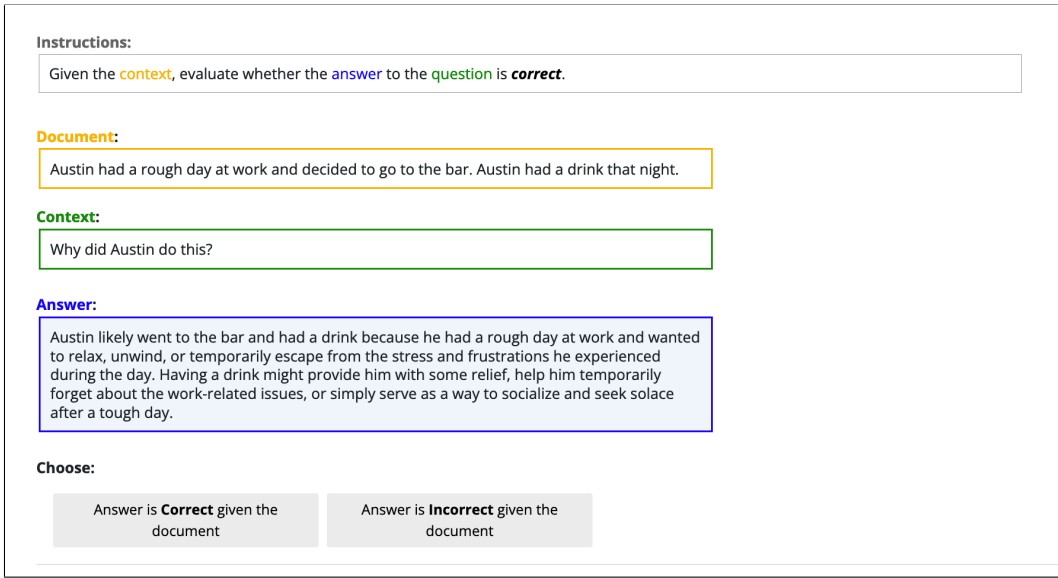

Figure 18: Model generation evaluation template.

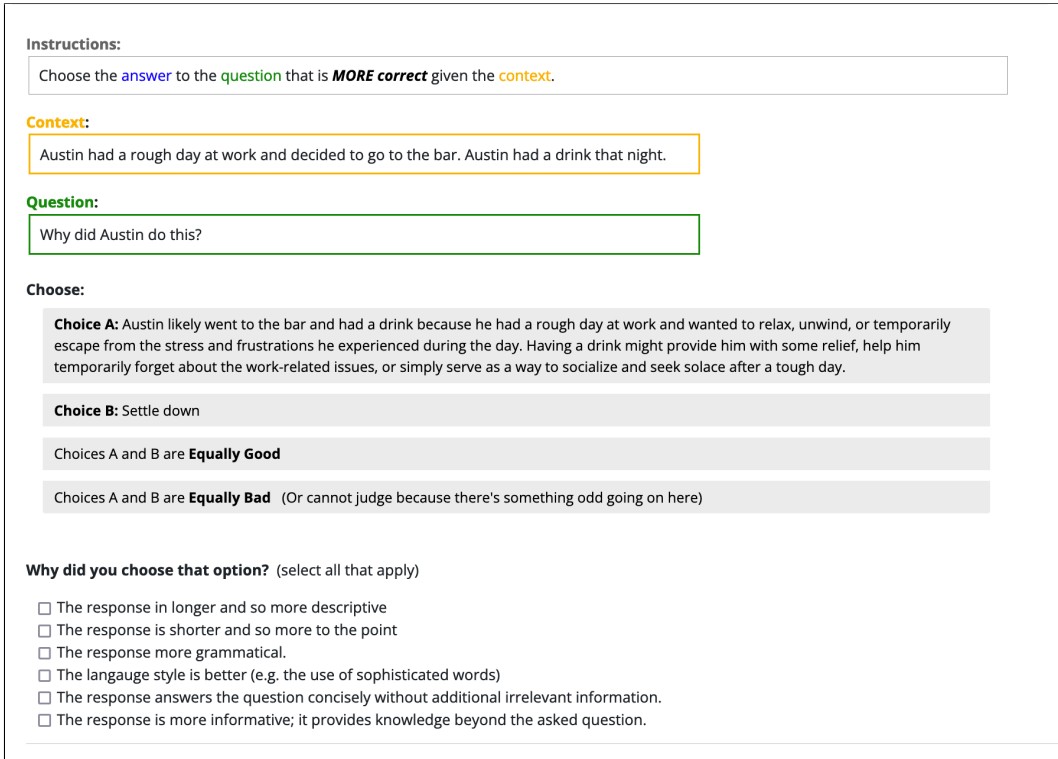

Figure 19: Comparative evaluation template.

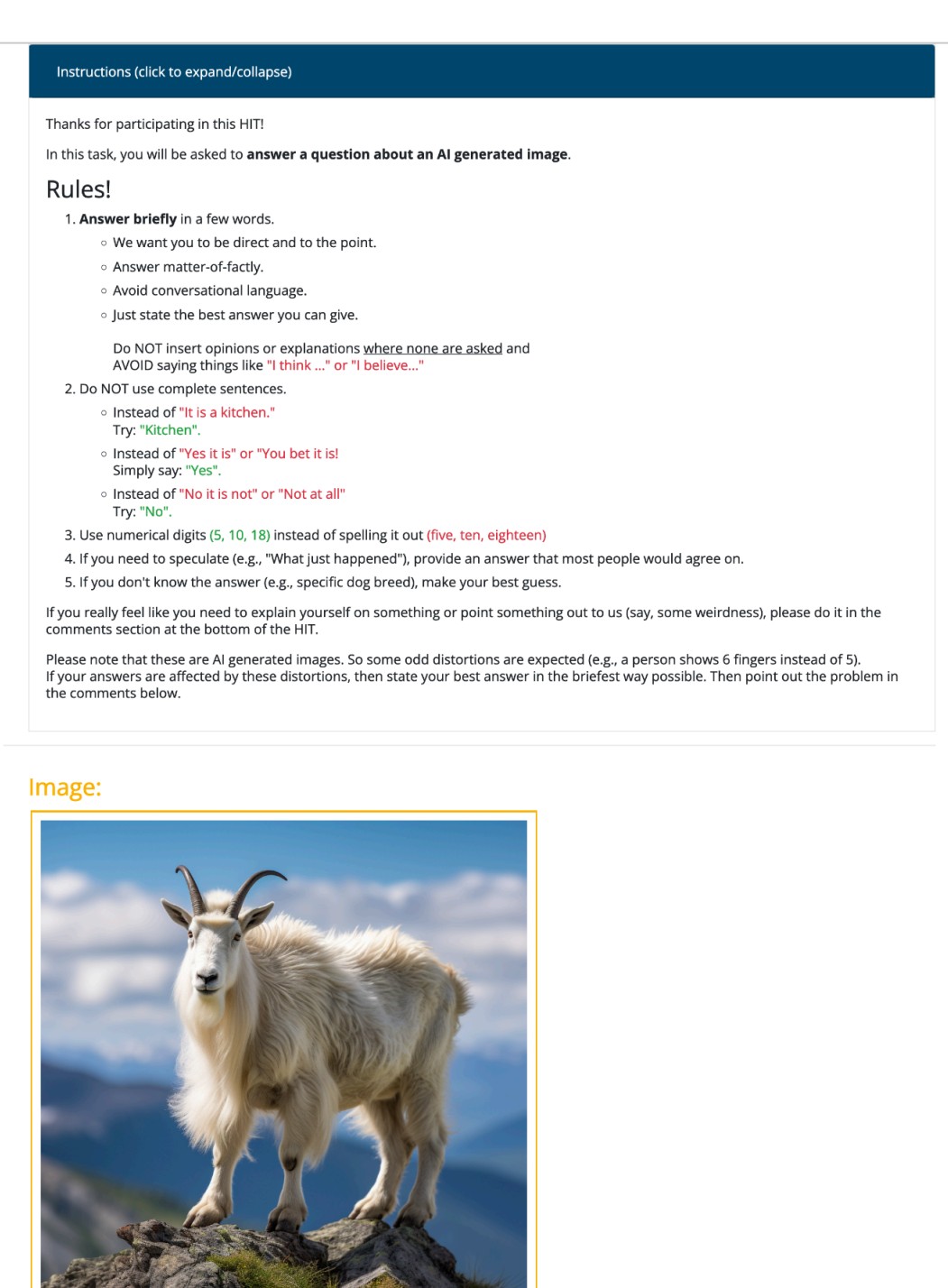

Figure 20: Human writing template

