# OpenReview forum: "The Generative AI Paradox: “What It Can Create, It May Not Understand”"
_ICLR.cc/2024/Conference — ICLR 2024 poster_

### Official Review · Reviewer_W7YK · 2023-10-22

**Soundness:** 2 fair
**Presentation:** 2 fair
**Contribution:** 3 good
**Rating:** 6
**Confidence:** 4

**Summary:**

Foundation models are growing increasingly powerful. However, their behaviour can be perplexing: often, it seems their ability to generate compelling outputs exceeds their ability to understand their own outputs. These authors put a name to this phenomenon: “the Generative AI Paradox.” The authors attempt to explore dimensions of this phenomenon by comparing human and model performance across two modalities (language and vision).

**Strengths:**

The motivation for the paper is superb. The authors do an excellent job introducing this paradox and it is nice that they put a name to the phenomenon. I am glad to see the authors studying this behaviour. In that sense, the conceptual underpinnings of the paper hold value for the broader ML community.

Further, I believe that the LLM results are especially compelling. Figure 2 is a nice empirical demonstration of the Paradox. In fact, I was particularly interested in Figure 10 in the Appendix; it is interesting what factors lead human raters to prefer GPT responses over humans. I’d encourage the others to move these findings up to the main text.

**Weaknesses:**

While I admire the authors’ motivation and their language-centric experiments, I believe that the results for the vision domain are fundamentally flawed. The Paradox hypothesis that the authors raise in Hyp 1 refers to the **same model** being equivalent in generative performance but worse in discriminative performance relative to humans; therefore, I think it is not experimentally sound to have **different** vision models for understanding vs. generation. An adequate investigation into the hypothesis requires having the same model (like the authors did in the language task; e.g., GPT tested on generative and discriminative tasks) rather than one model for generative and (more than one, but different) for discriminative. This discrepancy, I believe, invalidates any of the vision domain results.

However, I do not think that this weakness is fatal. I would encourage the authors to focus just on the language domain, unless they procure a vision (or language+vision?) model for which they can study generative and discriminative performance jointly in the same system. If the authors instead focus on just language, I would encourage them to move Fig 10 to the main text (per my note above). At this time, such a chance would be quite major, so I unfortunately recommend rejecting the work. Though, I think the core idea of this paradox, and the language results, warrant further study.

A smaller weakness: the authors are perhaps a bit too flippant about the use of the word “understanding” and “intelligence” from a human cognition perspective. I encourage the authors to look into Gardner’s Theory of Multiple Intelligences in particular. This weakness pales in comparison though to the urgency and weight of the first weakness raised.

More details on the human data used are needed as well (see below).

**Questions:**

- What data was used for human generations in the language domain? This was not clearly spelled out from my reading? Did you use the language data from the benchmarks discussed? If so, please provide more details.
- What version of GPT-4 did the authors use? The March 14 version? The “live” API instance? If the latter, were results conducted all in the same time window? Otherwise, I would worry about a silent update possibly impacting the discrepancy.
- Can you please provide further details on how participant agreement was calculated? Pairwise in what sense? You reference “kappa” in the Appendix footnote… is this Cohen’s Kappa? Can you provide more details on the skew noted?
- Minor note which did not impact my score: I would encourage the authors to break Section 2.1 and 2.2 into their own full sections (2 and 3, respectively).

---

> ### Author Response · Authors · 2023-11-22
>
> We thank reviewer W7YK for conducting a careful and comprehensive review, and for finding that **“The motivation for the paper is superb”**, and that the authors **“do an excellent job introducing this paradox”**.
> We note that key changes to the paper in the updated draft are colored red in the updated draft, and clarify points below with respect to the reviewer’s main concerns.
>
> ### **“I think it is not experimentally sound to have different vision models for understanding vs. generation”**
> We agree that a single vision model that both generates and understands would be an ideal system to test here. However, we make some clarifications to explain our decision:
>
> 1. Hypothesis 1 is referring to generative models as a technology rather than a single model in particular. That is, surprisingly good generation is achievable with generative models, compared to sometimes weak understanding that such models have achieved. We have worked to make this clearer in the paper text. (Section 2.1)
>
> 2. Vision models are capable of both image generation and understanding. Image generators can be used for understanding [1] but with significantly weaker performance than image understanding specialist models (which we use). Thus, our current results likely underestimate the extent of the generation-understanding discrepancy when considering a single model, as the generation performance will remain the same but understanding performance will be lower than what we have reported.
>
> Thank you for bringing up this ambiguity. We have now clarified this in the paper text.
> ### **“What data was used for human generations in the language domain? This was not clearly spelled out from my reading?”**
> We use the human-written reference responses from the existing datasets that we test on. As these tend to be extensively validated, the only risk here is of overestimating human performance which would underestimate the prevalence of the paradox. This has been clarified in the paper text  (Section 3 and Appendix D).
> ### **“Can you please provide further details on how participant agreement was calculated?”**
> There is a natural label bias in the annotations where many of the responses fell into a single label. In this case, standard inter-annotator agreement statistics are not reliable (the well-known paradoxes of kappa [2]). We thus reported the pairwise agreement, that is, percentage of the time two or more annotators agreed on a label. .This has been clarified in the appendix footnote 8.
> ### **“Can you provide more details on the skew noted?”**
> This is explained by the response above, with respect to label imbalance.
> ### **“the authors are perhaps a bit too flippant about the use of the word ‘understanding’ and ‘intelligence’ from a human cognition perspective.”**
> We use these terms following the conventions in the field, specifically considering commonly used terms like “Artificial Intelligence” and “Natural Language Understanding”. We have further clarified that these words are intended in a directed way, as when used in the given terms. (Section 1)
>
> ### **References**
> [1] Li, A. C., Prabhudesai, M., Duggal, S., Brown, E., & Pathak, D. (2023). Your diffusion model is secretly a zero-shot classifier. arXiv preprint arXiv:2303.16203.
>
> [2] Feinstein AR, Cicchetti DV. High agreement but low kappa: I. The problems of two paradoxes. J Clin Epidemiol. 1990;43(6):543-549. doi:10.1016/0895-4356(90)90158-l

---

> > ### Comment · Reviewer_W7YK · 2023-11-22
> > **Thanks for your response!**
> >
> > Dear Authors,
> >
> > Thank you for your response and clarifications. Many of my questions have been addressed.
> >
> > As for point 1 wrt the different vision models --- I appreciate the inclusion of footnote 2. However, I still find the specification of Hypothesis 1 and discourse a bit misleading. For instance, Hyp 1 implies that "model" is the same model that is passed to g() and u()? But in all vision experiments, these are different models. It could help to specify model_u, model_g explicitly and note that these are the same in the language case, but different for vision.
> >
> > I have increased my score in response to the other questions, and again, appreciate the excellent work the authors have done in the language domain! But still feel somewhat unsettled at this time with the way that the different-vision-models are employed and discussed. The specification of the Generative AI Paradox as the authors have it still leans heavily towards the single-model framing in my view.
> >
> > Again, thank you for taking the time to respond and update the text!

---

> > > ### Author Response · Authors · 2023-11-22
> > > **Addressing feedback on Hypothesis 1 and editing the paper**
> > >
> > > Dear Reviewer,
> > >
> > > Thank you for your thoughtful feedback and increased score. We appreciate your recognition of our work in the language domain.
> > >
> > > Regarding your concern about Hypothesis 1 and its discourse, we acknowledge the potential for confusion. To address this, we explicitly specified model_u and model_g in Equation 1 (Sec 2.1), making it clear that these can be different models for both generation and understanding (Sec 2.2.2 ). We further highlighted why we considered the same model for language versus two models for vision (Sec 2.2.2 ).
> > >
> > > Your feedback is invaluable, and we are committed to enhancing the clarity of our framing.
> > >
> > > Once again, we appreciate your time and consideration.
> > >
> > > Best,
> > > Authors

---

> > > > ### Comment · Reviewer_W7YK · 2023-11-23
> > > > **Adjusted score :)**
> > > >
> > > > Thank you, I appreciate your work to clarify on this matter. I think the change is helpful, and have increased my score accordingly!

---

> > > > > ### Author Response · Authors · 2023-11-23
> > > > > **Thank you :)**
> > > > >
> > > > > Dear Reviewer,
> > > > >
> > > > > Thank you, we appreciate the adjustment in your score. We're glad the changes were helpful. If you have any further questions, please feel free to let us know.
> > > > >
> > > > > Best,
> > > > > Authors

---

> > > > > > ### Comment · Reviewer_W7YK · 2023-11-23
> > > > > > **Future Suggestion :)**
> > > > > >
> > > > > > Great! All my questions are answered for now. I do want to note that I'd be *quite* excited to see the authors extend their paradigm and experiments to a vision/multimodal model that (in a single model) handles generation + understanding. As you note, perhaps the Paradox observed would be even stronger! It seems like some of the new GPT models could be a good fit... e.g., those which combine DALL-E generation with GPT-4-V ("understanding"). Just a suggestion for future work; not here, necessarily. I'd be keen to see the results regardless, and I wish the authors luck in their future research!

---

### Official Review · Reviewer_n1Ya · 2023-10-24

**Soundness:** 4 excellent
**Presentation:** 4 excellent
**Contribution:** 3 good
**Rating:** 8
**Confidence:** 4

**Summary:**

This work identifies and investigates the "Generative AI Paradox", where generative models (language and vision) can create outputs equal to (or beyond) that of human experts but do not understand said outputs. The underlying notion is that generation for humans is dependent on understanding (as a prerequisite for expert-level outputs), but the same is not true for generative AI. The hypothesis is tested through an assessment of generative abilities and understanding abilities (split into selective and interrogative settings), for both the language and vision domains. Results show gen. AI outperforms humans on generative tasks but underperform on understanding. The work goes on to discuss the possible reasons for this paradox occurring.

**Strengths:**

**Originality**
O1. The work demonstrates novelty through the formalisation of hypotheses to capture the paradox.
O2. Through evaluation with human participants, novel and concrete findings are established to test the proposed hypotheses.

**Quality**
Q1. Evaluation is conducted using several models and datasets for both the vision and language domains.
Q2. Thorough analysis shows support for the hypothesis, with a strong accompanying discussion.

**Clarity**
C1. Work is well presented and figures aid understanding.
C2. Additional results and further discussions are provided in the Appendix.
C3. Paper is easy to follow w.r.t to introducing the paradox, formalising it, presenting results, and then discussing findings.

**Significance**
S1. Further studies beyond initial hypothesis testing reveal additional results, such as human discrimination being more robust to challenging inputs.
S2. Some discussion of potential explanations as to why the paradox occurs, e.g. gen. AI trained on generative learning objective - understanding is only encouraged if it furthers this goal (a divergence from human learning).

**Weaknesses:**

**Clarity**
C1. Additional diagrams showing the steps in experiments would aid understanding. This would help highlight the difference between the experiments in Sections 3 and 4, where (from my understanding) the former uses existing candidates to evaluate discriminative understanding, and the latter uses generated outputs. Showing exactly where in the experimental process generative outputs vs existing data are used would be informative. Figure 1 doesn't quite capture the difference between the two cases in my opinion.
C2. For the discriminative vs generative subplot in Figure 2, it is difficult to see all of the blue points for GPT4. Adjusting the plot to use alpha values would allow these points to be seen and strengthen the claims supported by the plot. Similar for Figure 8 in the appendix.
C3. A handful of concrete outputs would provide further contextualisation for the kind of errors the models are making, as well as give further clarity to the structure of experiments. Figure 1 provides some examples, but additional outputs would be beneficial. Figures 12 to 15 help with this, but the questions/outputs are the same as used in Figure 1.

**Significance**
S1. While the discussion touches on possible explanations for the paradox, it does not mention ways it could be mitigated.
S2. The end of the abstract states "Our findings... call for caution in interpreting artificial intelligence by analogy to human intelligence". This is briefly discussed under broader implications in Section 7, but I think the findings warrant a more detailed discussion of this outcome and how the results should be used in future work.

**Questions:**

1. In relation to the S1 weakness above, are the authors able to suggest any ideas about how the paradox could be overcome? Potential explanations are provided, but how does understanding the paradox enable improved development and a reduction of the disparity between generation and understanding?
2. The authors note that Figure 2 shows sub-hypothesis 1 is supported for at least one model in 10/13 datasets. Do you have an understanding of why experiments on the other three datasets do not support it?

---

> ### Author Response · Authors · 2023-11-22
>
> We thank reviewer n1Ya for a careful review, and are happy that they find that **“novel and concrete findings are established”** with **“strong accompanying discussion”**, and that our **“work is well presented”**. We note that key changes to the paper in the updated draft are colored red, and we outline changes and clarifications below:
>
> ### **“Additional diagrams showing the steps”**
> We have added a diagram (Figure 8) to provide a more detailed explanation of the full process, thank you for bringing up this potential point of confusion! We are happy to elaborate on this.
>
> ### **“Adjusting the plot to use alpha values”**
> We have adjusted the alpha values in the plot in Figure 2, and have also slightly reduced marker size so that the full set of data points is more visible. We are happy to make further changes as the reviewer requests.
>
> ### **“A handful of concrete outputs would provide further contextualisation for the kind of errors the models are making”**
> We have added a set of qualitative examples to the appendix (see Appendix C.3) demonstrating the inability of language models to understand their own generations (interrogative), and general errors of vision models (both interrogative and selective understanding).
> ### **“it does not mention ways it could be mitigated.”**
> We have added ideas for possible mitigations to our discussion section. In short, we suggest alternative optimization objectives, reduction of model memorization, and broad incentivization of understanding capabilities as 3 paths forward, inspired by the causes that we propose. Understanding the cause of this phenomenon will be necessary to find a mitigation strategy.

---

> ### Comment · Reviewer_n1Ya · 2023-11-22
>
> Thank you for the rebuttal. The new figures and examples aid with clarity and understanding. Authors have addressed my concerns and answered my questions.

---

> > ### Author Response · Authors · 2023-11-23
> > **Thank you!**
> >
> > Dear Reviewer,
> >
> > Thank you, we're glad the new figures and examples were helpful. If you have any further questions, please feel free to let us know.
> >
> > Best,
> > Authors

---

### Official Review · Reviewer_icxt · 2023-10-31

**Soundness:** 4 excellent
**Presentation:** 2 fair
**Contribution:** 4 excellent
**Rating:** 6
**Confidence:** 3

**Summary:**

The paper introduces "the generative AI paradox," highlighting the disparity between generative models and humans, where the former excels in generation but lags in understanding. It substantiates this paradox through extensive experimentation, providing in-depth explanations for the observed phenomena. While "the generative AI paradox" resonates with the preconceived beliefs of many researchers, the paper uniquely confirms its validity, offers insightful explanations, and outlines potential avenues for future research, all of which serve as a significant source of inspiration for the broader community.

**Strengths:**

- The generative AI paradox is aligned with the intuitions of researchers, and is important to the community;
- The conducted experiments are comprehensive.
- This paper provides some future directions to explore.

**Weaknesses:**

- The insights presented in this paper may not be particularly surprising to the research community.
- (Please correct me if I've overlooked any details) The paper employs two metrics, selective and interrogative, to assess the understanding capability of generative models. However, it doesn't delve into which metric more accurately reflects the models' comprehension, nor does it discuss how each metric contributes to specific aspects of understanding. Additionally, a comparative analysis between the selective and interrogative abilities of the generative models is missing.
- The presentation, particularly the figures, requires further enhancement. Currently, they encapsulate extensive results without a clear presentation way, which somewhat complicates comprehension.

**Questions:**

- Could the authors explain which metric more accurately reflects the models' comprehension, nor how each metric contributes to specific aspects of understanding?
- Could the authors provide a comparative analysis of the generative models' selective and interrogative capabilities, and provide explanations for the observed results?
- The authors should consider enhancing the clarity and organization of the figures to improve the overall presentation.

---

> ### Author Response · Authors · 2023-11-22
>
> We thank reviewer icxt for carefully considering our  work, finding it **“important to the community”** and our experiments **“comprehensive”**.
> We note that key changes to the paper in the updated draft are colored red, and respond to the reviewer’s specific concerns below:
>
> ### **“employs two metrics, selective and interrogative … doesn't delve into which metric more accurately reflects the models' comprehension”**
> We agree that a single measure of understanding would simplify analysis. However, understanding is complex and has multiple distinct pieces, as demonstrated by the diverse ways that humans are tested for understanding (e.g. multiple choice questioning, long-answer, oral examination). Selective and interrogative evaluation in our experiments attempt to capture two core aspects of understanding but future work will need to explore more aspects, rather than collapsing onto one. We agree that a better description of what each evaluation is testing will be useful, and we have elaborated on this in the paper text (Section 2.1, definitions). In short, selective understanding concerns the ability to recognize correct answers, while interrogative concerns the ability to explain correct answers.
>
> ### **“... comparative analysis of the generative models' selective and interrogative capabilities, and provide explanations for the observed results?”**
> We study a different set of tasks for the two forms of understanding, especially in the language domain. We largely evaluate selective understanding on short-answer questions, where it is feasible to judge the correctness of an entire response (for human annotators), while we judge interrogative understanding on longer texts, in which it is more meaningful to understand the possibly complex information contained in the generation of the model. However, there are 3 tasks for which these evaluations overlap (XSUM, Mutual+, and HellaSwag) and so we have **added a comparison of these tasks to the Appendix** (Figure 14). The two forms of understanding seem to correlate, likely related to the relative difficulty of understanding the underlying task and stylistic aspects of the data. Understanding this relation warrants further study.
>
> ### **“insights presented in this paper may not be particularly surprising”**
> While we agree that our work matches the intuition of many researchers, much work recently relies on the understanding ability of generative models (e.g. model-based NLG evaluation [1]), or even makes this idea explicit. For instance, recent highly-cited work on GPT-4 states that “one of the key aspects of GPT-4’s intelligence is its generality, the ability to seemingly understand and connect any topic” [2]. We believe that work like ours is needed to provide more concrete, experimental proof of the ways in which models can lack understanding, and break from human intuitions.
>
> ### **“The authors should consider enhancing the clarity and organization of the figures”**
> We thank the reviewer for pointing out points of possible confusions, and have made some improvements to our presentation, including the addition of an experimental setup diagram (Figure 9), and improving the visibility of all data points in Figure 2 (bottom right). We are happy to incorporate any specific suggestions the reviewer has.
>
> ### **References**
> [1] Liu, Yang et al. “G-Eval: NLG Evaluation using GPT-4 with Better Human Alignment.” ArXiv abs/2303.16634 (2023): n. pag.
> [2] Bubeck, Sébastien et al. “Sparks of Artificial General Intelligence: Early experiments with GPT-4.” ArXiv abs/2303.12712 (2023): n. pag.

---

### Official Review · Reviewer_izog · 2023-11-10

**Soundness:** 3 good
**Presentation:** 4 excellent
**Contribution:** 3 good
**Rating:** 8
**Confidence:** 3

**Summary:**

The authors seek to study "understanding" in generative models. They conducted several experiments to examine the performance of generative models in both language and image domains in terms of generation vs understanding. It was found that the models surpassed human capabilities in generation tasks as already known but it was found that they consistently under-performed when it comes to understanding.

**Strengths:**

The work is timely, well written and significant for a large section of the conference's audience.
While it is a common observation that generative models seem to struggle with discrimination, this work studies it in a principled way while additionally collecting human data.
The work attempts to be complete as it covers a range of tasks, state of the art models and two modalities.

**Weaknesses:**

* The models tested were trained/finetuned for generation tasks. It is unclear if finetuning for discrimination will fix some issues if not all i.e does the performance drop come from not "understanding" or not from being familiar with discrimination tasks.

**Questions:**

* Did you collect any data which would reveal as to whether the participants can themselves figure if the models don't have understanding? That is, if you let participants interact (perhaps on a task where they need to collaborate) with the model (they are not told that it is a model), what percentage will complain that the model does not understand what it is outputting?

---

> ### Author Response · Authors · 2023-11-22
>
> We thank reviewer izog for their feedback. We are excited that reviewer izog stated our paper is “timely”,  “well written”, “principled” and “significant” for the research community and found our experiments, spanning a wide spectrum of tasks and two modalities, to provide a "complete" analysis.
> We note that key changes to the paper in the updated draft are colored red, and make the following clarifications:
>
> ### “Did you collect any data which would reveal as to whether the participants can themselves figure if the models don't have understanding?”
> While we don’t collect a large dataset of such examples, we have added our own interactions with GPT-4 to the appendix (C.3). Particularly, we include examples of simple questions which ask the model about its own generated stories, and the (often incorrect) answers that it outputs, to provide explicit examples of what the authors judge to be qualitative examples of misunderstanding.

---

### Author Response · Authors · 2023-11-22
**General Response**

We thank reviewers for careful and constructive feedback on our work, including finding it **“significant for a large section of the conference's audience”** (reviewer izog),  **“important to the community”** (reviewer icxt), with **“novel and concrete findings”** (n1Ya) and and finding the authors **“do an excellent job introducing this paradox”** (reviewer W7YK). We thank the area chair for carefully considering our work. We have made the following changes, clarifications, and improvements to our paper as part of the rebuttal, with changes colored red in the updated document:
- The addition of further examples of our evaluation to Appendix C.3.
- The addition of a diagram illustrating the overall experimental setup more clearly (Figure 8)
- Addition of a short discussion of potential mitigations to the Paradox of Generative AI in our discussion section.
- Changes to Figure 2 to increase clarity
- Additional clarifications in the paper text concerning the kind of understanding we are testing (Sec 2.1)

We believe that the vast majority of reviewer concerns have been addressed, either by changes to the paper draft or through clarification of paper details.

---

### Meta-Review · Area_Chair_MP3e · 2023-12-19

**Metareview:**

Modern models can produce outputs which they may not understand. This has been observed by others before, but has not been called out and systematically investigated before.

Reviewers found the paper to be interesting and the idea to be worthwhile. It is interesting that most humans cannot create output like image generation models, but can understand their output; while machines can produce such elaborate images, but often cannot understand them. Formalizing this idea and testing it is valuable, as future work can now investigate with this is (where during training it occurs, etc.). This may also have a significant impact on the construct validity of benchmarks as well as their design. Overall, this is of wide interest to the broader ICLR community.

**Justification For Why Not Higher Score:**

A relatively small number of tasks is used. It's unclear if this effect is decreased or increased as models are scaled up. This is critical as it makes all the difference between the effect being transitory and it potentially being a limiting factor.

**Justification For Why Not Lower Score:**

N/A

---

### Decision · Program_Chairs · 2024-01-16

Accept (poster)